# Timing of iceberg scours and massive ice-rafting events in the subtropical North Atlantic

Alan Condron ⬡ [1]✉ & Jenna C. Hill[2]

High resolution seafloor mapping shows extraordinary evidence that massive (>300 m thick) icebergs once drifted >5,000 km south along the eastern United States, with >700 iceberg scours now identified south of Cape Hatteras. Here we report on sediment cores collected from several buried scours that show multiple plow marks align with Heinrich Event 3 (H3), ~31,000 years ago. Numerical glacial iceberg simulations indicate that the transport of icebergs to these sites occurs during massive, but short-lived, periods of elevated meltwater discharge. Transport of icebergs to the subtropics, away from deep water formation sites, may explain why H3 was associated with only a modest increase in ice-rafting across the subpolar North Atlantic, and implies a complex relationship between freshwater forcing and climate change. Stratigraphy from subbottom data across the scour marks shows there are additional features that are both older and younger, and may align with other periods of elevated meltwater discharge.

[1] Geology and Geophysics, Woods Hole Oceanographic Institution, Woods Hole, MA, USA. [2] United States Geological Survey, Pacific Coastal & Marine Science Center, Santa Cruz, CA, USA. ✉email: acondron@whoi.edu

High resolution images of the sea floor from the western subtropical North Atlantic reveal over 700 individual iceberg scours spanning the southern US Atlantic margin, from Cape Hatteras, North Carolina (~35°N), to the Florida Keys (~24°N), in water depths from 170–380 m, that are traceable for >30 km (Fig. 1, Supplementary Fig. 1 and ref. [1]). The appearance of these features at such low latitudes is highly unexpected, not only because of the exceptionally high melt rates in this region (sea surface temperatures are 20–25 °C), but also because these features lie beneath the northward flowing Gulf Stream (Fig. 2). Indeed, in our prior work[1] the Gulf Stream in the glacial North Atlantic flows north along the continental shelf of North America until it detaches from the coast near Cape Hatteras, much like present day[2]. In the Mid-Atlantic Bight region to the north, cold subpolar slope waters flow south from the Grand Banks of Newfoundland until they encounter the Gulf Stream at Cape Hatteras (Fig. 2). Hence, for icebergs to reach the subtropical scour locations south of Cape Hatteras they must have drifted against the normal northward direction of flow over the continental slope; i.e., in the opposite direction to the Gulf Stream. The iceberg scours along the margin are thus interpreted to represent the plowing paths of iceberg keels transported more than 5000 km south along the United States continental margin to southern Florida in a cold, coastal boundary current derived from the former Laurentide Ice Sheet (LIS; refs. [1,3]).

The discovery of icebergs in this location has direct implications for understanding cryosphere–ocean–climate interactions as it suggests a narrow, buoyant, coastal boundary current must

have flowed from the Northern Hemisphere ice sheets directly to the subtropical North Atlantic gyre (~20°N–40°N) and that south of Cape Hatteras this current was moving in the opposite direction to the northward flowing Gulf Stream at depth. Research over the last 30 years has repeatedly shown that increases in freshwater (icebergs/meltwater) discharge to the *subpolar* North Atlantic can weaken the strength of the Atlantic meridional overturning circulation (AMOC) on multidecadal-to-millennial timescales by reducing North Atlantic Deepwater (NADW) formation[4,5].

The presence of iceberg scours in the subtropics confirms there must have been periods when a significant fraction of icebergs and meltwater released from the east coast of North America were routed directly to the subtropical North Atlantic gyre, bypassing regions of deep-water formation that are thought to regulate the AMOC[1,6]. While this freshwater is eventually advected northward by the Gulf Stream, turbulent mixing would have caused the water to be considerably saltier by the time it reached the subpolar gyre, making it less efficient at weakening deep-water formation and reducing the strength of the AMOC, compared to freshwater discharged directly to the subpolar gyre[1]. This routing and mixing of freshwater thus implies that the influence of meltwater on global climate may be more complex than previously thought. Understanding the timing and circulation of meltwater and icebergs through the global oceans during glacial periods is therefore vital for unraveling how past changes in high-latitude freshwater forcing influenced shifts in climate.

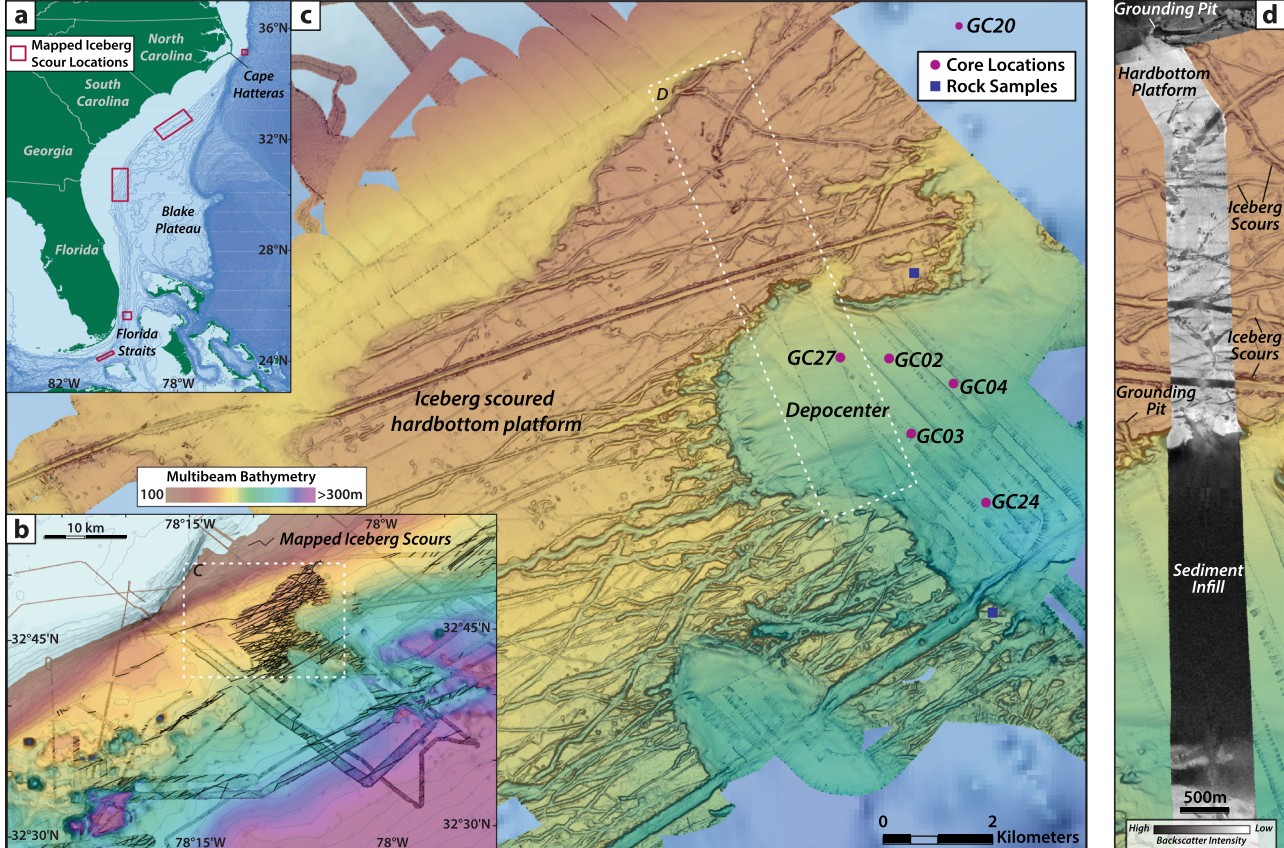

**Fig. 1 High resolution seafloor bathymetry and core locations. a** Seafloor iceberg scours have been mapped between 170–380 m water depth in several locations, shown by red boxes, where sufficient multibeam bathymetry data exist (ref. [1]). **b** Seafloor bathymetry of the iceberg scour site offshore of South Carolina, where ~500 individual iceberg scours have been mapped in the existing multibeam bathymetry. **c** Sediment cores used in this study were collected from buried iceberg scours in a depocenter adjacent to the iceberg scoured hardbottom platform. **d** Backscatter data across the study area indicate a rocky, hardbottom substrate on the iceberg scoured platform, with sediment infill in the local depocenter.

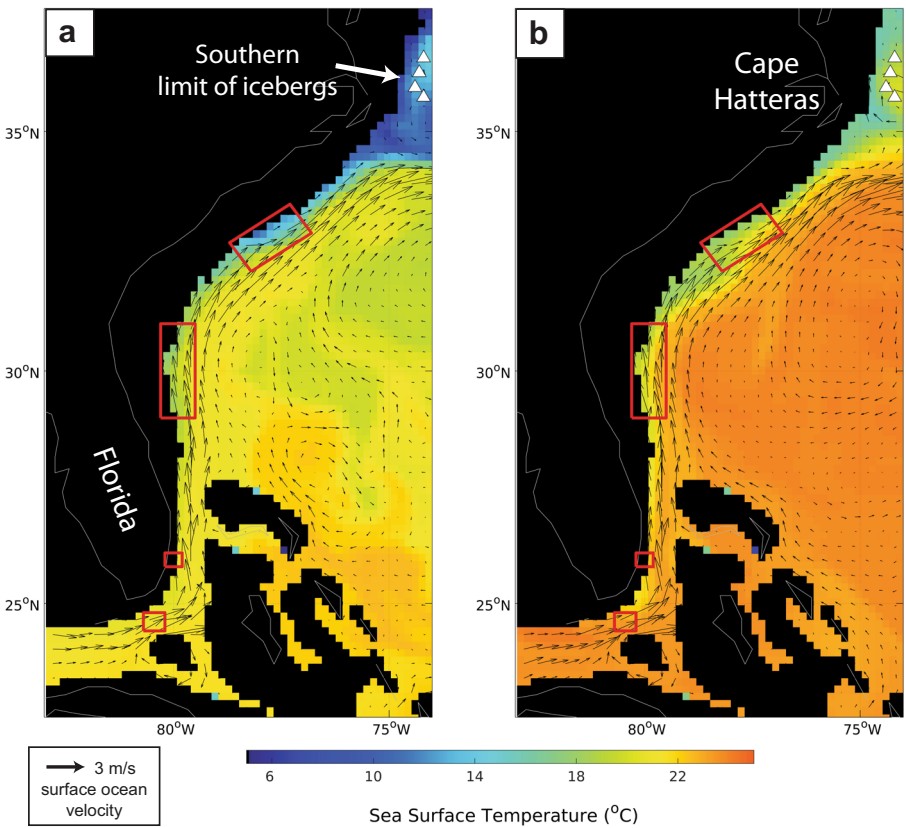

**Fig. 2 Simulated glacial sea surface temperature and surface ocean velocity.** The plots are drawn for January (**a**) and September (**b**) to show that the relict iceberg scours are located beneath the northward flowing Gulf Stream where ocean temperatures in our glacial ocean circulation model simulations are >20 °C. White triangles above 35°N represent the maximum southerly location that icebergs are able to drift to in our control simulations (i.e., no meltwater forcing) and highlight that icebergs do not freely drift to the scour sites in the absence of meltwater currents. Black arrows denote the strength and direction of the surface ocean flow.

Here we report on the sedimentology and ages of several buried iceberg scour marks observed along the subtropical US continental margin, south of Cape Hatteras. An iceberg model is then used to determine the mechanisms that led to the formation of these features. We conclude by considering the implications of our results for understanding the factors controlling the patterns of ice-rafted debris (IRD) across the subpolar North Atlantic (i.e., the IRD-belt) and the role that meltwater input to the ocean plays in modulating deep-water formation and large-scale ocean circulation.

## Results

**Sedimentology and ages.** To ascertain the age of the subtropical iceberg transport events, large diameter gravity cores were collected from sediment filled iceberg scours on the upper slope offshore of South Carolina (Fig. 1; ~33°N; 78°W ~200 m water depth). The buried iceberg scours were identified in Chirp sub-bottom profiles as small-scale, v-shaped incisions that occur along regular surfaces[7,8] within a small depocenter, adjacent to a large, iceberg scoured, hardbottom platform (Fig. 1, 3). The subbottom data show multiple erosional surfaces comprised of nested scours (Fig. 3), which is indicative of large numbers of icebergs and repeated iceberg scouring events. Typical iceberg scour incisions in this region are several meters deep, consistent in size with surficial iceberg plow marks observed in the multibeam bathymetry[9].

Sharp erosional contacts within the cores, along with abrupt changes in sediment character, correspond with erosional iceberg scour surfaces identified in the subbottom data. The erosional scour surfaces are overlain by sharp increases in grain size (>80% coarse fraction) and angularity, decreased sorting, and much greater abundance of glauconite, phosphorite, carbonate, and shell hash (Fig. 3). Rock and sediment samples from the nearby hardbottom platform show a similar composition that suggests local provenance of this coarser material (Fig. 1c; refs. [10,11]). Icebergs grinding southward across the hardbottom platform may have generated debris that was subsequently flushed into the adjacent depocenter by the reintroduction of the northward flowing Gulf Stream. The lithology of these local inputs makes it difficult to distinguish IRD here on the basis of grain size or carbonate content; however, several samples show a slight increase in abundance of angular quartz grains (>150 μm) around the basal scour surface that could be an indication of IRD, similar to deeper sites nearby (e.g., refs. [12,13]).

Accelerator mass spectrometer (AMS) $^{14}$C dates acquired from the most pristine *Globigerinoides ruber* species sampled above and below the scour surfaces indicate multiple scouring events between ~26.3–39.8 kyr (Fig. 4 and Supplementary Table 1). Three of the cores (GC02, GC04, and GC27) show an erosional surface with ages that cluster around ~31,000 calendar years BP (Fig. 4), which is roughly synchronous with Heinrich Event 3 (H3; ref. [14]). Core GC24, collected from a deeper, isolated depocenter where no scours were observed, also shows a significant increase in coarse material deposition around ~31 ka that persists through the top of the core (Fig. 5). The basal scour surfaces in cores GC02 and GC27 appear to be older, between

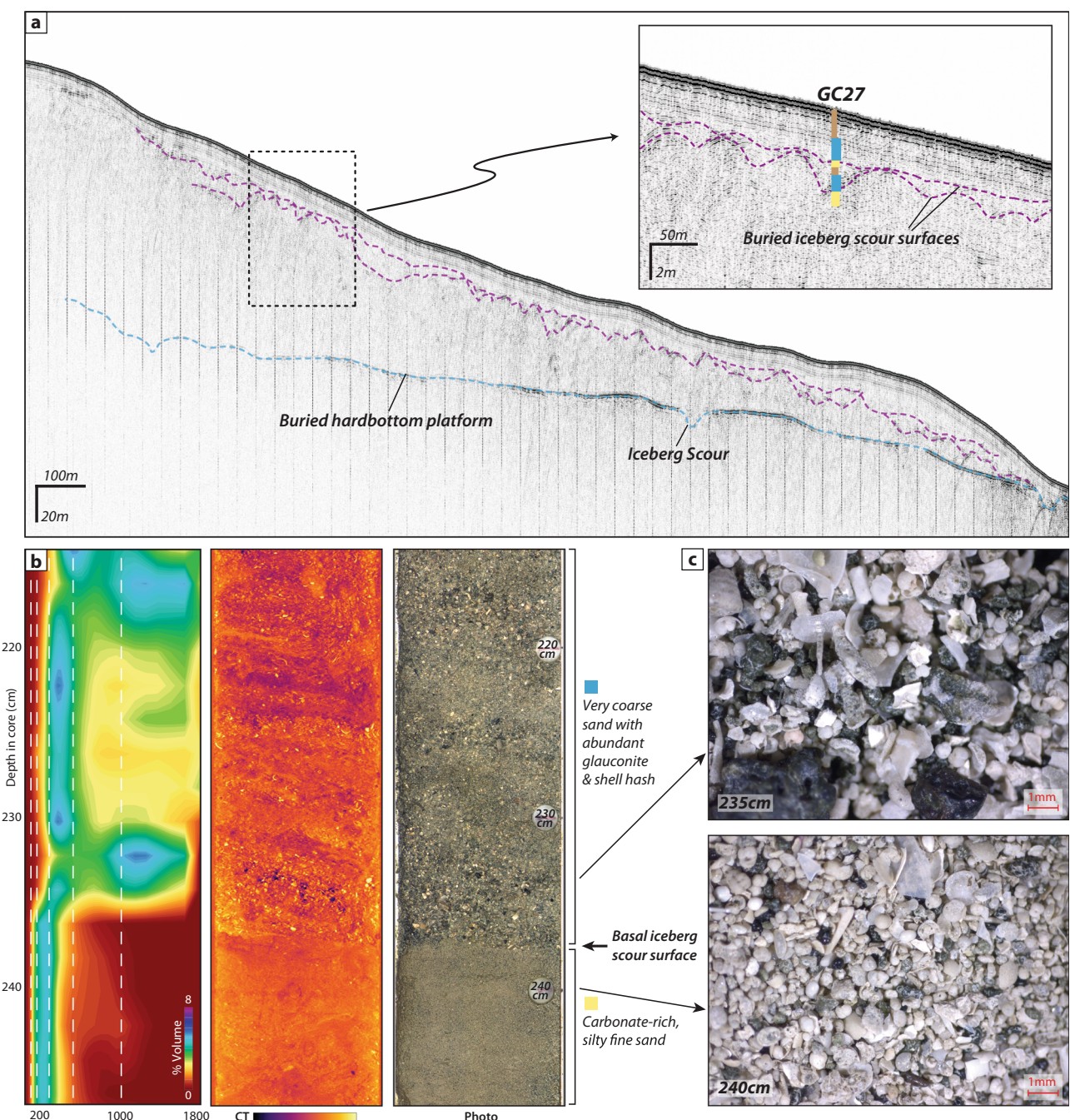

**Fig. 3 Example of buried iceberg scour sedimentology and stratigraphy. A** Chirp subbottom profile across the depocenter where core GC27 was collected. This area shows multiple nested iceberg scour surfaces within the sediment package, as well as some older scours that appear to have cut into the underlying hardbottom platform. **B** Example of lithological changes associated with the basal iceberg scour surface observed in GC27, as indicated by changes in grain size, density, and sediment composition. **C** Microscope photos of the coarse fraction indicate distinct lithologies, where coarse sand and gravel with abundant glauconite and shell hash are found above the basal iceberg scour surface, in contrast with finer, carbonate-rich sand below.

32–37 ka and <39 ka, respectively (Fig. 4). Additional scour surfaces observed in the subbottom data appear stratigraphically older than those sampled, including some that cut into the buried hardbottom platform (Fig. 3). Some of the cores record possible more recent iceberg scour surfaces (e.g., GC02 (~28 ka) and GC04 (~26 ka); Fig. 4); the erosional contacts are not as sharp here, but show distinct changes in sediment character at these times. Together, these results suggest there were at least 3–4 iceberg scouring events reaching subtropical latitudes. This is also consistent with observations of both seafloor and buried iceberg

scours along the New Jersey margin (~39.5°N) where regional stratigraphic correlations have been used to suggest there may have been four periods of southward iceberg transport at that location, roughly correlated with Heinrich Events 1–4[15].

We suspect that our limited sampling, with short (less than 3 m) cores targeted at sites where the scour surface shoaled, may have introduced a bias toward younger events, as well as shallower tracks from smaller icebergs, providing only a snapshot of subtropical iceberg transport events. The upper meter of all the cores typically consists of substantially bioturbated and reworked

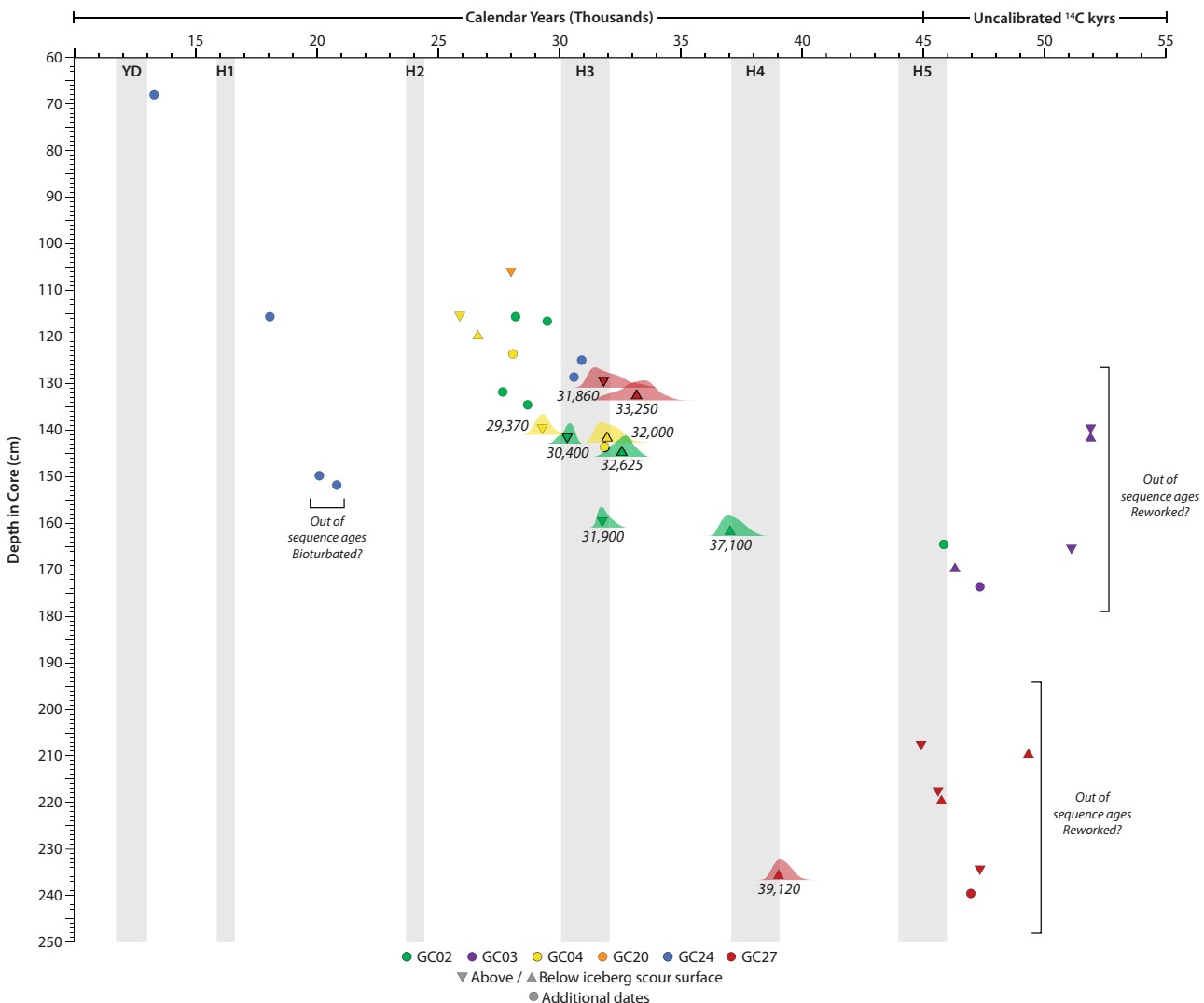

**Fig. 4 Radiocarbon dates from sediment cores collected in and around buried iceberg scours.** Median calendar ages, derived from radiocarbon calibration using Oxcal 4.3 are reported for samples younger than 45 kyr [14]C BP, while older samples are reported uncalibrated. Several of the cores show an erosional iceberg scour surface with ages above and below the surface that cluster around ~31 kyr cal BP. Several major climatic events (e.g., Younger Dryas, Heinrich Events) are highlighted with gray bars.

material that may obscure any possible evidence of scours in this section. The apparent absence of more recent sediment in the cores, based on the lack of *Globorotalia menardii*, a foraminifera species that was absent from the North Atlantic until early Holocene[16], and pre-Holocene dates, also suggests the heavy weight on the coring device may have resulted in overpenetration, such that the most recent sediment layers were not sampled. Alternatively, there may have been limited deposition in the Holocene. Both scenarios leave open the intriguing possibility of younger iceberg scour events that were not recorded in the samples from this location.

**Iceberg modeling**. To address how these icebergs reached subtropical latitudes, we developed a dynamic-thermodynamic iceberg model and coupled it to the Massachusetts Institute of Technology General Circulation Model (MITgcm;[17]) ocean–sea ice model (See Methods). All of our simulations were conducted using an eddy-permitting horizontal ocean grid resolution of 1/6° (~18 km) that is capable of resolving narrow coastal meltwater currents along the shelf and large-scale eddies (see ref. [1]). These

coupled ocean–sea ice–iceberg model simulations thus mark a significant step forward in paleoclimate modeling as they are the first-time glacial iceberg discharge events have been simulated at such a high spatial resolution.

In brief, rates of iceberg melt are based on mass loss from sensible heating, incoming solar radiation, wave erosion, and buoyant vertical convection. The horizontal drift of each iceberg in the model is then calculated from the sum of the drag forces exerted on the ice by the wind, ocean, sea ice, Coriolis force, and sea-surface slope. To account for changes in horizontal ocean velocity with depth, a novel multilevel keel drag scheme—similar to those used in state-of-the-art, short-term (2–3 days) iceberg forecasting—was employed (Supplementary Fig. 2; ref. [18]). Here, the net ocean drag on each iceberg is derived by summing the drag force exerted at each vertical ocean model level the iceberg keel penetrates. The inclusion of this scheme is found to be extremely important for simulating iceberg drift south of Cape Hatteras where meltwater from the LIS is moving south at the surface and in the opposite direction to the northward flowing Gulf Stream at depth. In this region, the lower part of an iceberg's keel can penetrate into the Gulf Stream waters to oppose

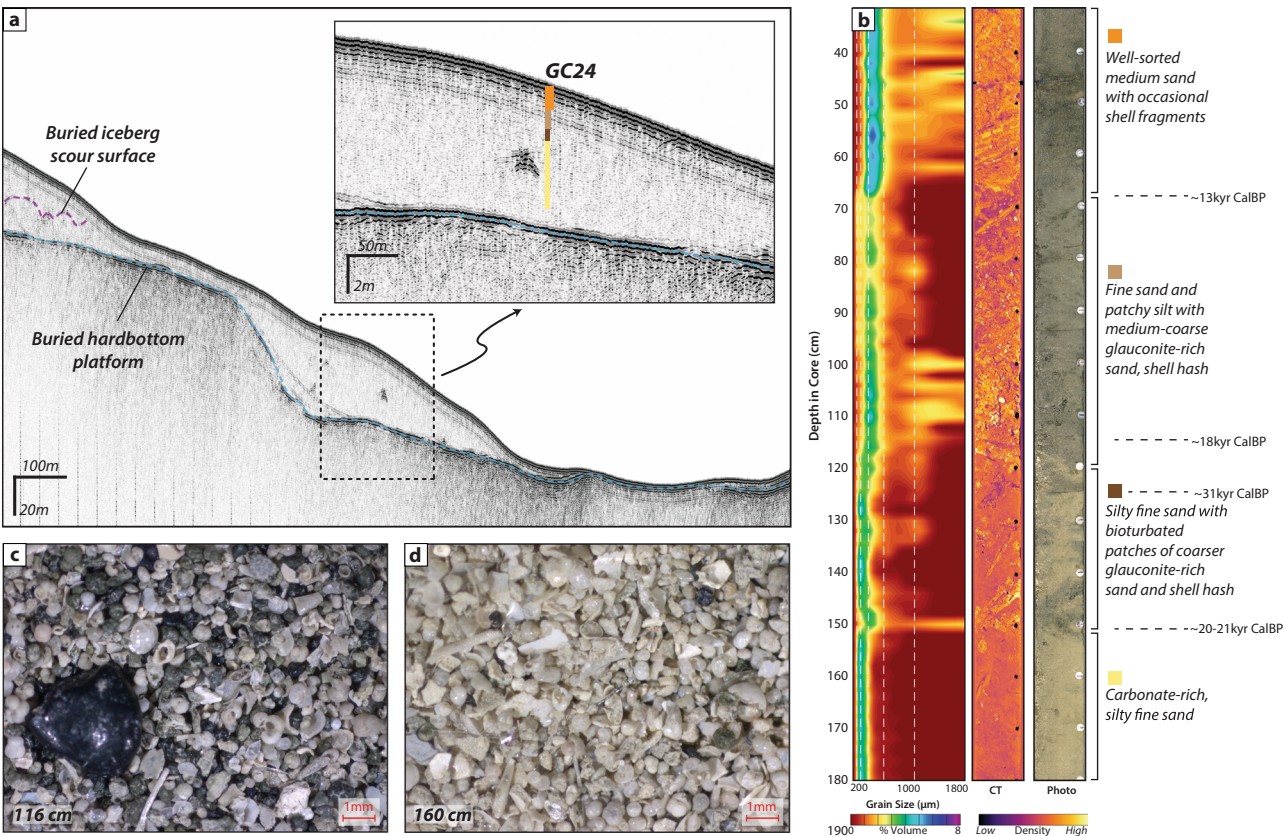

**Fig. 5 Sedimentology and stratigraphy of nearby unscoured region.** Core GC24, collected in a deeper portion of the depocenter, where no iceberg scours are observed, also shows distinct variations in grain size and lithology that correspond in time to the changes observed in the cores collected from within iceberg scours. **a** Chirp subbottom profile across the core location; **b** Grain size, false-color CT scan, and core photograph of the middle section of GC24; **c**, **d** Microscope photographs of coarse fraction (>63 μm) samples showing coarser grains with increased glauconite that occur after ~31 kyr cal BP, relative to the older carbonate-rich fine sands.

the southward drift and constrain the number and size of icebergs reaching the scour sites.

Finally, we developed a novel technique to simulate iceberg plow marks on the sea floor by allowing iceberg keels to penetrate up to 20 m into the seafloor sediment before becoming grounded and stationary. Once an iceberg grounds on the seafloor, it then remains immobile until it melts sufficiently to refloat and start drifting again. A full model accounting for the bottom drag caused by icebergs plowing the sea floor was considered too complicated at this stage given it would need to account for both the rheology of the marine sediment and the precise shape of the iceberg keel below the waterline, but we consider this approach to be a good first approximation given that most of the observed scours are incised up to 20 m deep into the sea floor sediment[2].

In each experiment, 6300 Gt yr$^{-1}$ (~0.2 Sv; Sv = $10^6$ m$^3$ s$^{-1}$) of ice is calved from three locations close to Hudson Bay, Canada, to reflect both known iceberg source regions and estimates of ice discharge during Heinrich Events[14,19]. In the control simulation (without any meltwater flood), icebergs from Hudson Bay drift south in the Labrador Current and then across the northern North Atlantic, as far east as the Iberian Margin (Fig. 6). In agreement with marine sediments containing IRD deposited during major Heinrich Events[14,20], the highest concentrations of icebergs are found in the subpolar gyre, between latitude bands ~40°N–50°N. Icebergs also drift along the continental margin as far south as Cape Hatteras (35°N, 74°W) to where the southward flowing shelf and slope waters meet the ~2 m/s northward flow of the Gulf Stream. The meeting of the slope waters with the Gulf Stream then inhibits any further southward iceberg movement

and the ability of icebergs to freely drift to any of the relict plow marks observed on the sea floor.

To explore the relationship between the northward flow of the Gulf Stream and the southward flow of the slope water in controlling the southern limit of iceberg drift, an additional model experiment was performed with the wind field over the North Atlantic shifted south to artificially push the Gulf Stream south (See Methods). In response to this change in wind forcing, the Gulf Stream detached ~1° further south of Cape Hatteras, compared to the control simulation, and allowed icebergs to freely drift to the most northern relict scour sites off the coast of South Carolina. Significantly, this is also where the greatest number of plow marks have previously been identified on the sea floor (ref. [1]), suggesting that the precise latitude at which the Gulf Stream separates from the coast in a glacial ocean[21] controls the ability of icebergs to reach the most northern scour locations. South of this region, however, the persistent northward flow of the Gulf Stream continued to inhibit icebergs from reaching the scour sites located off the coast of Florida (Supplementary Fig. 3). A different forcing mechanism—rather than a change in the position and/or detachment of the Gulf Stream from the coast—is thus needed to explain the occurrence of most of the scour features.

To explicitly examine the mechanisms capable of transporting icebergs to the most southerly scour sites, meltwater with fluxes of 2.5 and 5 Sv was released from Hudson Bay, Canada, to reflect prior research showing that the meltwater flux must be ≥2.5 Sv to form a narrow coastal current capable of reaching the most southerly scour sites[1]. In all of our meltwater flood experiments, entirely fresh (0 psu) water was released over an area of ~130 km$^2$

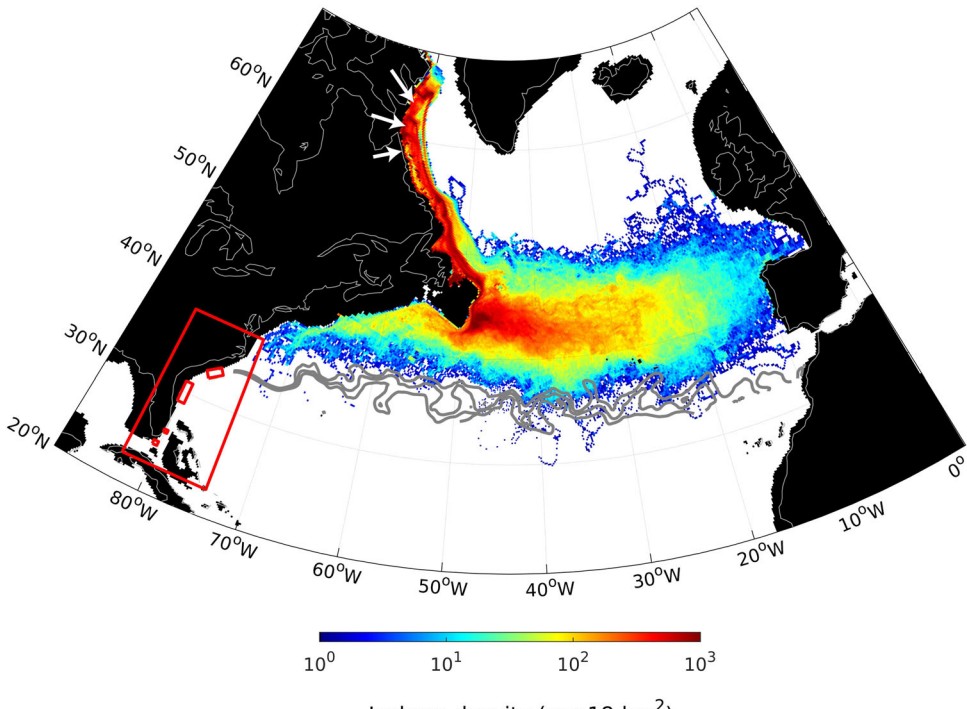

**Fig. 6 Simulated distribution of icebergs in the glacial North Atlantic.** In general, icebergs are restricted to the subpolar North Atlantic (40°N–50°N) where high concentrations of ice-rafted debris are found in glacial marine sediments (refs. [13, 19]). Icebergs do not freely drift to the relict subtropical scour sites, south of Cape Hatteras (small red boxes). The position of the Gulf Stream is marked by the 13–15 °C isotherms at 200 m water depth (gray contour lines); iceberg calving margins near Hudson Bay are denoted by the white arrows, glacial landmasses are shown in black, and the modern coastline by the gray line. The large red box highlights the regions displayed in detail in Fig. 7e, f.

at the surface of the ocean model (into the four model grid points closest to the drainage outlet) for 1 year (starting January 01) to simulate the rapid drainage of a large proglacial lake to a new level. Reconstructions of the volumes of freshwater released to the ocean during these outburst events are poorly known, but they are estimated to have peaked at 5 Sv during the 8.2 kyr event[22]. The time taken for a lake to lower to its new outlet is also uncertain, although hydrologic modeling estimates suggest that these events may have only lasted for up to 1 year[23]. Note also that the 0.2 Sv flux of icebergs calved from the Hudson Bay region is applied constantly throughout the model simulations; i.e., prior to the release of meltwater, during the 1-year meltwater outburst floods, and after the meltwater floods have ceased.

In both experiments, icebergs rapidly (~1–2 m/s) drifted southward in the Labrador Current and reached the Grand Banks of Newfoundland after ~15 days. Icebergs then continued to drift in a south-southwest direction along the east coast of North America, reaching the latitude of Nova Scotia (~44°N), ~3200 km from Hudson Bay after 40 days (Fig. 7). As the meltwater continued south of Cape Hatteras, hundreds of icebergs were able to drift towards the most northern relict iceberg scour sites off the coast of South Carolina. The ability of the meltwater to continue to flow south of Cape Hatteras then depends on the magnitude of the flood given that the ice-laden coastal flow is essentially a buoyant gravity current. Consistent with theoretical and laboratory studies of buoyant gravity currents along a sloping bottom in a rotating fluid[24], the meltwater is observable in the model as a bulge in sea surface height (SSH) with larger floods producing currents that are (vertically) thicker and extend farther offshore (Supplementary Fig. 4). Note also that the ability of our model to capture the vertical structure and flow of these currents is implicit on using a "free-surface height" scheme and that they would not be resolved

in models using a more traditional "rigid-lid" approach to study changes in meltwater input on climate. Our results show that if the SSH of the meltwater is larger than the SSH of the Gulf Stream at Cape Hatteras then the meltwater will continue to flow south beyond this point and, in our model, this is the case for both the 2.5 and 5 Sv outburst floods, but not for smaller events (Supplementary Fig. 4).

In our experiment releasing 2.5 Sv of meltwater, icebergs were only able to drift to South Carolina, despite the coastal current propagating through Florida Strait and into the Gulf of Mexico (Supplementary Fig. 5). An inspection of the change in horizontal ocean velocity (with depth) in this region indicates that the meltwater current becomes very shallow (upper 10–20 m) in this region and that the Gulf Stream continues to flow northwards below this. As such, the drag force exerted on the upper part of each iceberg keel from the southward flowing meltwater is insufficient to overcome the force of the Gulf Stream acting on the lower part of the keel. Again, this highlights the requirement to use a multilevel keel drag scheme in the iceberg advection routine to accurately simulate iceberg transport to the scour sites.

When the meltwater flux was increased to 5 Sv, icebergs continued drifting south of South Carolina, such that 120 individual icebergs passed through Florida Strait (26°N) (Fig. 7). In this region, the meltwater was confined to the western side of the strait, with a width of ~40 km and southward velocity of 1–2 m/s, down to ~60 m in the water column (Supplementary Fig. 6). In addition to being relatively fresh, the coastal meltwater is also exceptionally cold (~5–8 °C), compared to the surrounding offshore waters (~20–25 °C), as a result of limited entrainment and mixing with the ambient subtropical ocean (Fig. 8). The persistence of this cold current thus helps to reduce melt rates as the icebergs move south from the cold subpolar region to the much warmer subtropical western North Atlantic.

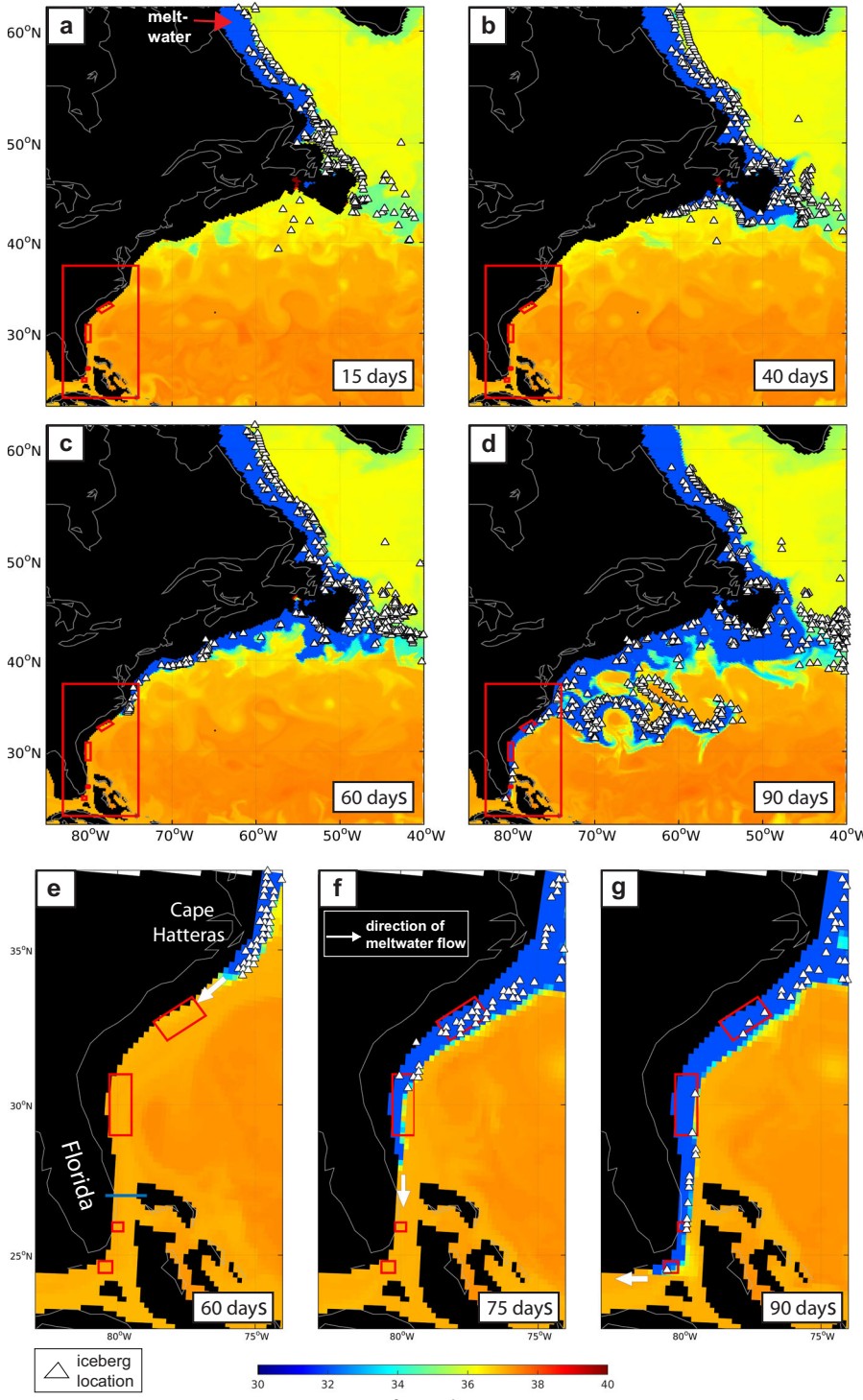

**Fig. 7 Simulated influence of elevated meltwater discharge on iceberg drift.** Hundreds of icebergs (denoted by the white triangles) entrained in the glacial meltwater drift southwards along the continual shelf (**a**, **b**) reaching Cape Hatteras after 60 days (**c**, **e**). After 75 days, icebergs reach the relict scour sites off South Carolina and Northern Florida (**f**), and continue south through Florida Strait to the most southerly scours after 90 days (**d**, **g**).

As the meltwater continues to be discharged from Hudson Bay, the current persists, creating a remarkable ~6100 km southward flowing "conduit" along the entire east coast of North America, from Hudson Bay to the subtropics, that allows additional icebergs to drift over the scour sites. In other words, it seems that the iceberg scours off the coast of Florida are a record of truly massive outburst flood events. In addition, while our experiments only consider the transport of icebergs calved from the Hudson Bay region, it is entirely possible that icebergs originating from more southerly parts of the LIS and/or far-field calving margins such as Greenland and Iceland could have been "swept along" with the meltwater (provided they were south of the drainage outlet at the time of the flood) and contributed to the formation of the subtropical iceberg plow marks.

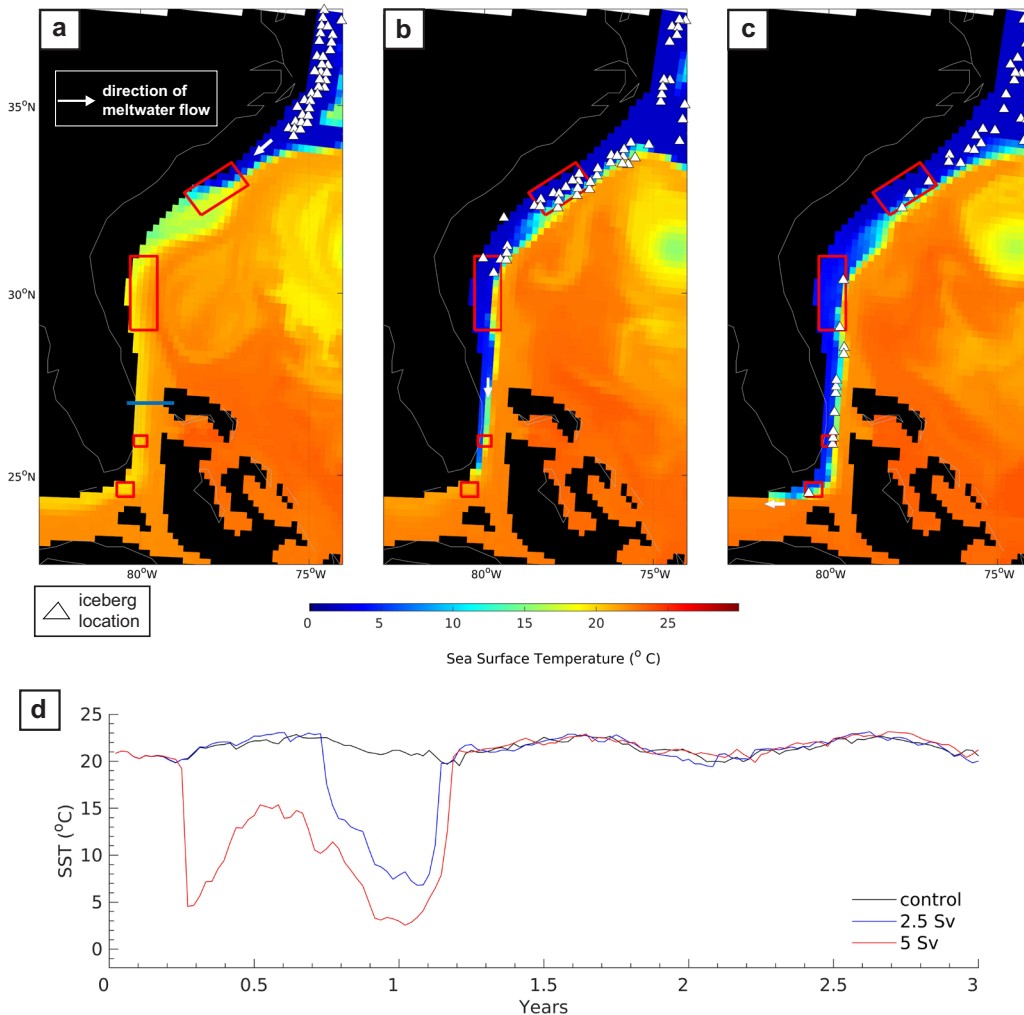

**Fig. 8 Sea surface temperature in response to elevated meltwater discharge from Hudson Bay, Canada.** Snapshots of sea surface temperature (**a–c**) 60, 75, and 90 days after a 5 Sv meltwater flood was released from Hudson Bay, Canada, and correspond to the same time periods shown in panels e–g of Fig. 7. The blue line at Florida Strait (in panel **a**) highlights the cross section used to compile the time series of sea surface temperature show in panel **d**. Iceberg locations are denoted by the white triangles.

By allowing iceberg keels to plow through the sediment on the continental shelf in the model, we found that scouring occurs in the same geographical regions as the plow marks observed in the high resolution multibeam imagery (Fig. 9). Consistent with the observations, the number of keel marks decreases in abundance moving south along the margin, with ~200 plow marks simulated off the coast of South Carolina, compared to only ten at Florida Keys. The modeled scours at South Carolina (~32.5°N) are also oriented in a similar south-southwest direction (~189°) to the observations (198–206°) and lie in comparable modern-day water depths (142–256 m in the model versus 170–380 m in the observations (Fig. 9c)). An examination of the total number of icebergs drifting south of Cape Hatteras, compared to the number of scours, also reveals that only ~5–25% of icebergs scour the sea floor. Hence, the number of icebergs reaching the subtropical western North Atlantic Ocean would likely have been much larger than the number of scours implies.

In addition to the transport of icebergs to the relict scour sites, our model shows that ~10,600 icebergs (~15–20% of the total number in the North Atlantic) were transported offshore, into the subtropical gyre, to the south of the main IRD-belt (Fig. 10 and Supplementary Fig. 7). Icebergs also reached Bermuda Rise (32° N, 65°W) in the Sargasso Sea, where IRD has been reported in marine sediment cores[12,13]. While the presence of IRD at this

location has previously been explained by the entrainment of icebergs in cold core rings helping to reduce ice melt and allowing them to cross the Gulf Stream, our findings present a second mechanism by which icebergs reached this destination.

The model indicates that the appearance of icebergs at subtropical latitudes in the western North Atlantic would have been dependent on the existence of the coastal meltwater current, as icebergs are quickly reconfined to the subpolar gyre once the elevated levels of meltwater are reduced, and the coastal meltwater current disappears (Fig. 10 and Supplementary Fig. 7). Indeed, Fig. 10 indicates that at the onset of the meltwater event, icebergs are primarily restricted to the region 40°N–50°N, as also shown in Fig. 6. However, after 1 year of elevated meltwater discharge the geographical distribution of icebergs has significantly expanded to include much of the subtropical western North Atlantic, such that icebergs are advected southeast toward the Bermuda Rise. Once the meltwater discharge is reduced though, icebergs become restricted to the subpolar North Atlantic even though the freshwater signature of the meltwater persists in the subtropics (Fig. 10c).

## Discussion

Our analysis of marine sediments indicates that icebergs drifted south to subtropical regions multiple times during the last

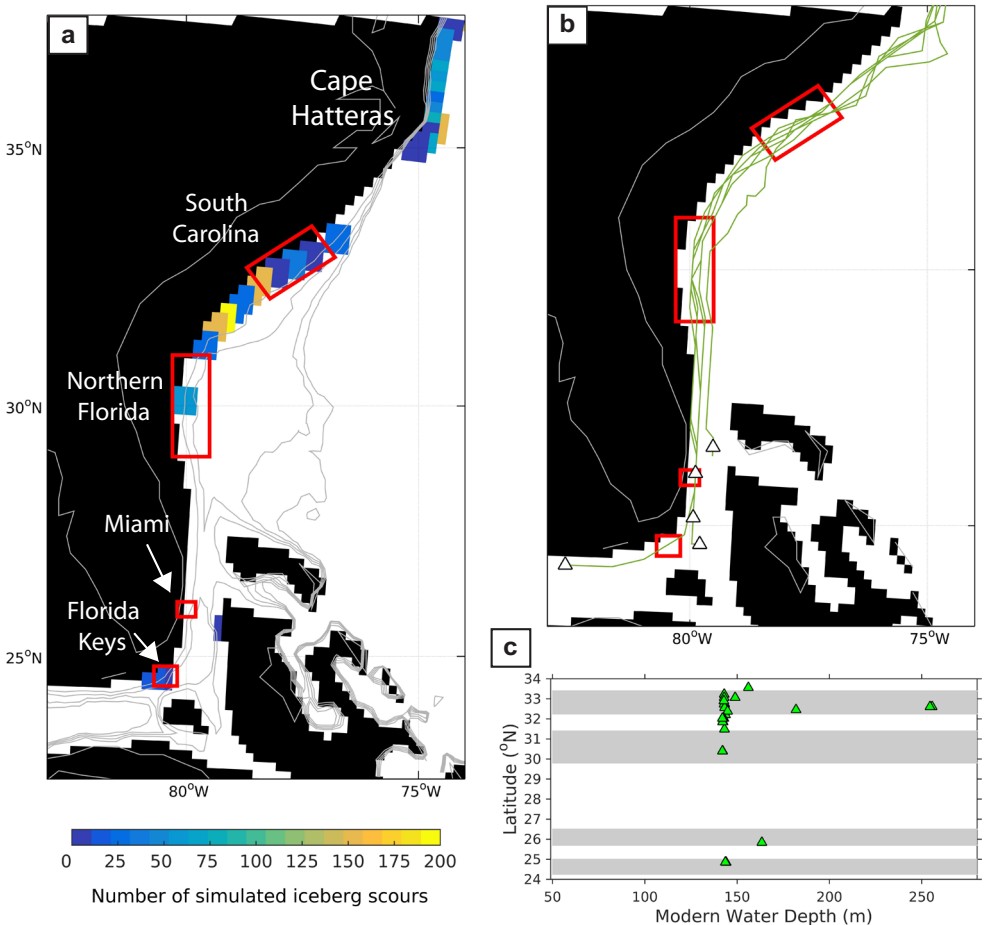

**Fig. 9 Simulated iceberg scours along the east coast of the United States. a** The number of iceberg scours simulated in the model, south of Cape Hatteras, in response to 5 Sv of meltwater; red boxes are locations where relict iceberg scours have been observed using multibeam bathymetry data. **b** Drift trajectories (green lines) and the maximum southerly locations (tringles) of icebergs scouring the sea floor, **c** Distribution of simulated iceberg scour depths with latitude (filled green triangles); gray shading corresponds to the four observed scour locations as reported in ref. [1].

glaciation. While the age relationship does not imply a causative process, ~31,000 calendar years BP coincides with the period of massive iceberg discharge, Heinrich Event 3. Previous work indicates that H3 has several features that set this event apart from four of the six Heinrich layers (H1, H2, H4, and H5) that occurred in the last 60 kyr (e.g., refs. [14,25,26]). In particular, while IRD from Hudson Bay is found in the western North Atlantic during H3, it is significantly lacking in the eastern North Atlantic sector compared to these four H-events. In fact, IRD in the eastern North Atlantic appears to have been sourced from the Greenland and/or the Eurasian ice sheets during H3[14,25]. Heinrich Event 6 also shows a similarly modest increase in IRD and possibly a different IRD source, compared to the other four major events[14,26]. Gwiazda et al., (ref. [26]) proposed that this variation in IRD deposition during H3 reflected a greater confinement of icebergs sourced from the LIS to the western North Atlantic, but precisely why this might have been the case remains unknown. Here, we postulate that the repeat transport of icebergs to the western subtropical North Atlantic by large meltwater floods could explain this pattern, especially if such events occurred multiple times during H3.

We also note that a more southerly position of the Gulf Stream could have, in part, contributed to the observed change in IRD during H3 given that our model predicts an increased confinement of icebergs to the western North Atlantic when the wind field was perturbed (Supplementary Fig. 3). Given uncertainties in the concentration and partitioning of IRD within glacial icebergs

(e.g., ref. [27]), we also cannot rule out the possibility that a lack of IRD deposition during Heinrich Event 3 simply reflects a change in the concentration of IRD in the icebergs and/or a change in where the IRD is partitioned within the ice at this time. Indeed, "clean" icebergs with little or no IRD—analogous to modern-day icebergs calved from large ice shelves fringing Antarctic—would leave little or no IRD "fingerprint" on the sea floor, while icebergs with IRD concentrated in the basal portion of the ice would cause IRD to be deposited much closer to the calving margin.

Our high resolution coupled ocean–sea ice–iceberg model results indicate that ≥2.5 Sv of meltwater discharge from Hudson Strait is required to transport icebergs to the relict scour sites. This is higher than previous estimates for Heinrich Events (0.02–1 Sv; ref. [19]); yet these prior calculations are based solely on persistent ice rafting across the polar and subpolar regions and do not account for short-lived coastal boundary flows that appear to have periodically brought large volumes of ice-laden meltwater into the subtropics. We thus speculate that H3 and H6 could have actually been larger meltwater discharge events than the other H-events that carried icebergs south of the classic IRD-belt (40°N–50°N).

The alignment of high concentrations of scouring on the sea floor in our iceberg model in the same geographical regions as the observations confirms that the identified features are indeed relict iceberg scour marks caused by massive ice rafting events capable of reaching the subtropical western North Atlantic Ocean. The model also shows that icebergs can be advected to other

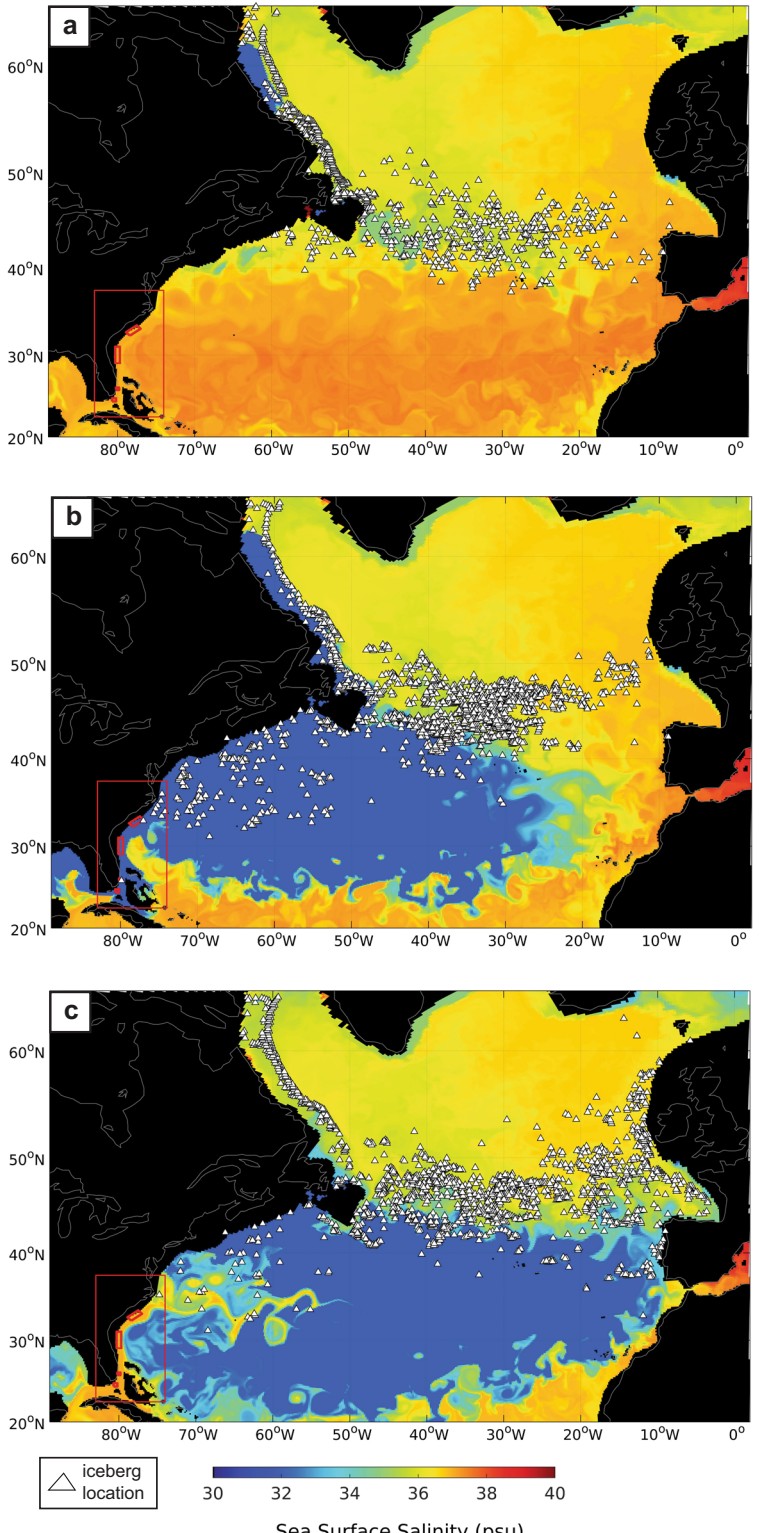

**Fig. 10 Simulated sea surface salinity and the distributions of icebergs in the glacial North Atlantic.** The top panel (**a**) shows that at the onset of the meltwater event, icebergs (denoted by the white filled triangles) are primarily restricted to the region 40°N–50°N, where high concentrations of ice-rafted debris (IRD) are found in marine sediments. After 1 year of elevated meltwater discharge (panel **b**), the geographical distribution of icebergs has expanded to include the subtropical North Atlantic. Once the meltwater discharge is reduced (panel **c**—drawn 1 year after the meltwater flood ended), icebergs become restricted to the subpolar North Atlantic even though the freshwater signature of the meltwater persists in the subtropical gyre.

subtropical sites (e.g., Bermuda and Bahamas) without invoking cold core rings that cross the Gulf Stream wall much farther north. Our findings thus demonstrate that the geographical region of the ocean influenced by meltwater freshening was not confined to the subpolar gyre but is consistent with previous studies[1,6] showing that the release of large volumes of iceberg-laden meltwater from Hudson Bay, Canada, leads to a significant freshening of the subtropical North Atlantic gyre (Fig. 10). This freshwater then undergoes significant mixing and is gradually advected northwards by the Gulf Stream towards the subpolar gyre. As a result, the freshwater is much saltier (less fresh) by the time it reaches high-latitude regions of deep-water formation (that likely modulate AMOC strength) than if it had been directly released to the subpolar gyre. This result is in contrast to both the notion that subpolar regions of deep-water formation were rapidly freshened by large outburst floods and the "classic" technique in numerical models of applying a uniform layer of freshwater to the subpolar North Atlantic (between 50–70°N) to study the impact of freshwater on AMOC and climate (refs. [4,5]). We postulate that the initial transport of significant volumes of freshwater to the subtropical North Atlantic as a result of massive glacial outburst floods, followed by the subsequent mixing of this water with the ambient ocean en route to the subpolar gyre, could explain the muted reduction in AMOC strength during Heinrich Event 3[28] given that meltwater would be saltier by the time it reached the subpolar gyre, and thus less capable of inhibiting deep-water formation.

The ages and stratigraphy of the scours discussed here suggest that there were multiple subtropical iceberg scouring events, consistent with observations from farther north along the New Jersey margin[15]. The iceberg scour ages presented here are also only from a subset of plow marks at the South Carolina site, and the future recovery of additional sediment cores from this location, as well as from the more southerly scours, will help reconstruct the timing and frequency of these events and determine whether they coincide with other Heinrich events.

## Methods

**Iceberg model**. The iceberg model is coded in parallel FORTRAN and is capable of simulating the melt and drift of 10,000's of icebergs in the ocean. Icebergs are assumed to be rectangular, with a width (W) to length (L) ratio of 1:1.62[29,30]. To clarify some terminology: The subaqueous part of the iceberg is referred to as the keel and the keel's thickness as draft (D); the aerial portion is known as the sail and the sails height above the sea surface as freeboard (Fb). In the model, the keel thickness and Fb height are derived from knowing the total iceberg thickness and the ratio of the density of ice to seawater. The equations used to derive iceberg drift and deterioration in the iceberg model are described in detail in Savage (ref. [31]) as well as below: Individual icebergs are simulated as Lagrangian particles, with their horizontal acceleration (units: m/s$^2$) calculated from the equation of motion for an iceberg:

$$m\frac{d\vec{v}}{dt} = -mf\hat{z}\times\vec{v} + \vec{F}_a + \vec{F}_w + \vec{F}_s + \vec{F}_p \quad (1)$$

where $m$ is the mass of the iceberg, $\vec{v}$ is iceberg velocity, $t$ is time, and the five terms on the right-hand-side represent the various forces (in kg/m/s$^2$) exerted on each iceberg: the Coriolis force $-mf\hat{z}\times\vec{v}$, where $f$ is the Coriolis parameter and $\hat{z}$ is the vertical unit vector, wind drag, $\vec{F}_a$, water drag, $\vec{F}_w$, sea ice drag, $\vec{F}_s$, and the horizontal pressure gradient, $\vec{F}_p$. The drag force from the wind is generated on both the vertical side walls of the iceberg above the waterline (form drag; $C_{av}$) and the horizontal surface plane (skin drag; $C_{ah}$) as:

$$\vec{F}_a = \left(\frac{1}{2}\rho_a C_{av} A_{av} + \rho_a C_{ah} A_{ah}\right)|\vec{v}_a - \vec{v}|(\vec{v}_a - \vec{v}) \quad (2)$$

where $\rho_a$ is air density, $\vec{v}_a$ surface wind velocity, $A_{av}$ and $A_{ah}$ are the vertical and horizontal cross-sectional areas of the iceberg (Supplementary Table S2). The drag force from the ocean accounts for changes in horizontal ocean velocity with depth by summing the drag force at each vertical ocean model level an iceberg's keel penetrates, based on Turnball et al., (ref. [18]), as:

$$\vec{F}_w = \sum_{i=1}^{n}\left\{\frac{1}{2}\rho_w C_{wv} A_{wv}(i)|\vec{v}_w(i) - \vec{v}|(\vec{v}_w(i) - \vec{v})\right\} + \rho_w C_{wh} A_{wh}(n)|\vec{v}_w(n) - \vec{v}|(\vec{v}_w(n) - \vec{v}) \quad (3)$$

where $i$ is the vertical ocean model level, $\vec{v}_w(i)$ is the water velocity at each vertical model level, $A_{wv}(i)$ and $A_{wh}(n)$ are the vertical and horizontal cross-sectional areas of the iceberg at each model level and at the base of the iceberg, and parameters $C_{wv}$ and $C_{wh}$ are the vertical form drag and horizontal skin drag coefficients, respectively. The drag force exerted by sea ice acts on the sidewalls of the iceberg and only on the part of the keel that is in the surface level of the model:

$$\vec{F}_s = \frac{1}{2}\rho_s C_{sv} L_\perp T_s|\vec{v}_s - \vec{v}|(\vec{v}_s - \vec{v}) \quad (4)$$

where $C_{sv}$ is the sea ice form drag coefficient, $L_\perp$ is the length of the iceberg normal to the stressing force at the surface level (i.e., width or length), $T_s$ is sea ice thickness, and $\vec{v}_s$ is sea ice velocity. The drag force is only considered when the concentration of sea ice exceeds 15%, while in high (>90%) concentrations of sea ice, icebergs drift with the pack ice (i.e., $\vec{v} = \vec{v}_s$) (ref. [32]). Finally, the pressure gradient force is calculated directly from the SSH, $\eta$, of the ocean model's nonlinear free surface as:

$$\vec{F}_p = -mg\vec{\nabla}\eta \quad (5)$$

Iceberg deterioration (units: m/s) is from solar radiation, sensible heating, wave erosion, and buoyant vertical convection. Freshwater from melting icebergs is released into the surface level of the ocean model with a temperature and salinity of 0 °C and 0 psu, respectively. Melt from solar radiation, $M_r$, reduces iceberg thickness as:

$$M_r = \frac{F_{sol}}{\rho_i \Gamma_i}(1 - \alpha) \quad (6)$$

where $F_{sol}$ is the solar radiation flux (W/m$^2$) derived from the local downward and shortwave radiation flux, $\Gamma_i$ is the latent heat of fusion of ice (J/kg), and α is the iceberg albedo (Supplementary Table 3). Subaerial melt from sensible heating (also referred to as forced convection), $M_{fa}$, is generated by the relative motion of the air passing the iceberg, and leads to both a reduction in waterline length and vertical thickness as:

$$M_{fa} = \frac{q_f}{\rho_i \Gamma_i} \quad (7)$$

where $q_f$ is the heat flux per unit surface area (W/m$^2$),

$$q_f = Nuk_a\triangle T/L \quad (8)$$

and $k_a$ is the thermal conductivity of the fluid, $\triangle T$ is the difference between the local air temperature and the iceberg($\triangle T = T_a - T_i$). The Nusselt number, $Nu$, gives the ratio of convective to conductive heat transfer as:

$$Nu = 0.055Re^{0.8}Pr^{0.4} \quad (9)$$

where the Reynolds number, Re, and Prandtl number, Pr, are defined as

$$Re = |\vec{v} - \vec{v}_a|L/\upsilon_a$$

$$Pr = \upsilon_a/D_a \quad (10)$$

where $\upsilon_a$ and $D_a$ are the kinematic viscosity and thermal diffusivity of air, respectively. Melt is also generated by sensible heating below the waterline, $M_{fw}$, and is calculated by replacing the constants for thermal conductivity, kinematic viscosity, and thermal diffusivity in Eqs. 8 and 10 with those for water (Supplementary Table 3). Iceberg melt below the waterline from buoyant vertical convection, $M_l$, along the sidewalls reduces an icebergs width and length as follows:

$$M_l = 8.82\times 10^{-8}\triangle T + 1.5\times 10^{-8}\triangle T^2 \quad (11)$$

where $\triangle T$ is the difference between the ocean water temperature and the freezing point temperature of seawater. Finally, iceberg melt from wave erosion, $M_w$, is simulated as:

$$M_w = 0.000146\left(\frac{R}{a}\right)^{0.2}\left(\frac{a}{W_p}\right)\triangle T \quad (12)$$

where $R$ is the roughness height of the iceberg and $W_p$ the wave period (Supplementary Table 3). The wave amplitude, $a$, is empirically related to wind speed and dependent on both sea ice fractional area and freeboard height, $Fb$, to avoid producing erroneously large wave drag forces. Finally, icebergs are considered to become unstable and roll-over when their length to thickness ratio is less than 0.7, ($L/T < 0.7$), and in this case, $L$ and $T$ are instantaneously swapped (ref. [30]).

The model uses ten iceberg size classes (Supplementary Table 4) that represent a modern-day iceberg distribution and are similar to those used in Bigg et al., (ref. [33]). Given uncertainties in the size of icebergs associated with Heinrich Events we consider this to be a reasonable first estimate. Moreover, as Fig. 9 (main text) shows that iceberg scouring in our model occurs in roughly the same water depths as the observations, our choice of iceberg size classes must closely approximate the size of actual icebergs drifting south of Cape Hatteras.

**Ocean model**. All numerical model simulations were performed using the Massachusetts Institute of Technology General Circulation Model (MITgcm) (ref. [17]). Our model configuration has an eddy-permitting horizontal global grid resolution of 1/6° (~18-km) with 50-levels in the vertical with spacing set from ~10 m in the

near-surface to ~450 m at a depth of ~6000 m. Ocean tracer transport equations are solved using a seventh-order monotonicity preserving advection scheme. There is no explicit horizontal diffusion, and vertical mixing follows the K-Profile Parameterization. Sea ice is simulated using a dynamic-thermodynamic sea ice model that assumes a viscous-plastic ice rheology and computes ice thickness, ice concentration, and snow cover[34].

The simulations were integrated under glacial boundary conditions: sea-level is 120 m lower than modern-day and the atmospheric boundary conditions (10-m wind, 2-m air temperature, surface humidity, downward longwave and shortwave radiation, precipitation, and runoff) are provided from output from the fully coupled Community Climate System Model version 3 (CCSM3) LGM integration[35]. The model was integrated forward using a 600 s timestep with the iceberg advection routine cycled ten times for every ocean timestep using a second-order Runge–Kutta method.

**Gulf Stream perturbation experiment**. To explicitly examine the sensitivity of southward iceberg transport to the point at which the Gulf Stream detaches from the coast at Cape Hatteras we performed an additional experiment in which the wind field (U, V) over the North Atlantic region (5–90°N; bounded on the east and west sides by landmasses) was displaced 5°S. As this shift leaves a gap in the wind field from 85–90°N, values in this region were simply replaced with the original values over this region.

**Sediment cores**. Large diameter gravity cores were collecting aboard the R/V Hugh R. Sharp in August 2017, using the Oregon State University (OSU) coring facility. The cores were logged for physical properties using a Geotek multi-sensor core logging system (Any use of trade, firm, or product names is for descriptive purposes only and does not imply endorsement by the US Government). Computed tomography (CT) scans were conducted on selected for cores using both the OSU medical system and the higher resolution USGS Geotek RXCT system. Several cores were also sampled every 2 cm for grain size. The grain size analyses were conducted at Coastal Carolina University, using a Beckman-Coulter LS13320 Laser Diffraction Particle Size Analyzer. Radiocarbon dates on foraminifera (*G. ruber*) were acquired from samples at key intervals within the cores and analyzed at the National Ocean Sciences Accelerator Mass Spectrometer (NOSAMS) and Beta Analytic, Inc. facilities.

## Data availability
The full set of equations describing the iceberg model's dynamics and thermodynamics are given in the Methods Section. The AMS $^{14}$C dates, iceberg model parameters, and the iceberg model-generated data, including the iceberg locations [Figs. 6, 7,10], sea surface salinity [Figs. 7,10], water temperature [Fig. 8], and iceberg scour locations [Fig. 9]) are all available in the Supplementary Material. The AMS $^{14}$C dates are also archived at https://www.ncdc.noaa.gov/paleo/study/33352. Any additional results from the simulations are available from the corresponding author upon request.

## Code availability
The MITgcm numerical ocean model code is available (mitgcm.org). The iceberg model code is available from the corresponding author upon request.

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

## Acknowledgements
We thank Summer Praetorious (USGS), Jerry McManus (Lamont), and Lloyd Keigwin (WHOI) for helpful comments and advice on an earlier version of this manuscript and faculty and students at Oregon State University and Coastal Carolina University for assistance with the sediment core collection and analyses. A.C.'s research was supported by the National Science Foundation Office of Polar Programs through NSF grant OCE-1903427 and the Biological and Environmental Research (BER) division of the US Department of Energy through grant DE-SC0019263. The numerical simulations were carried out using MITgcm on the Woods Hole Oceanographic Institution HPC machine, Poseidon. J.H.'s research was supported by the National Science Foundation Marine

Geology and Geophysics Program through NSF grant 1558994 and by the US Geological Survey Coastal and Marine Hazards and Resources Program.

## Author contributions
A.C. designed the iceberg model and performed all numerical modeling experiments. J.H. led the sediment core collection and analyses. A.C. and J.H. jointly wrote the manuscript.

## Competing interests
The authors declare no competing interests.

## Additional information

**Peer review information** *Nature Communications* thanks John Goff, Sidney Hemming, Robert Marsh, and other, anonymous, reviewers for their contributions to the peer review o this work. Peer review reports are available.

