## [Peer Review File · Nature Communications]

REVIEWER COMMENTS

Reviewer #1 (Remarks to the Author):

General Comments:

As a follow-up to Hill and Condron (2014), this paper presents new and critical sedimentary evidence for the timing of ice berg scouring, as well as new modeling that incorporates iceberg dynamics and thermodynamics into their circulation model. Although the latter part of the work is not my area of expertise, it appears to me to be a very significant modification to the ocean circulation modeling that provides remarkable insight into the transport of icebergs, either during Heinrich events or otherwise. Both components of the paper provide compelling results; each would have merited its own publication. But the results are also highly complementary and support each other, and combining the two in a single publication allows for a stronger statement of implications than could be made separately. Overall I found this an excellent paper, well written, and should be of interest to a wide audience given both the geological and climatological significance.

I don't know what the space limitations are for Nature Communications, but I do hope the authors can expand their figures shown in the main body of the paper. Figure 1 is, unfortunately, a particularly poor figure because too much was crammed into it, such that much of the important information and annotation were unreadable without significant magnification of the image. It is really 3 figures – panels A-D, panel E, and panel F. And instead of Figure 1E, I would hope the authors could move supplementary figure S3 into the main body of the text. Figure S3 is an excellent figure that best highlights the key findings of the sedimentary component of the paper. The chirp data are particularly important in confirming the existence of multiple episodes of iceberg scouring, even if all of them cannot be clearly dated. But that information, combined with the modeling work, provides a very compelling rationale for concluding that these >>must<< have been Heinrich events, because otherwise there is no way to get icebergs to this location. I was surprised, for example, that the authors did not make note in the text of the scour identified in the hard-bottom substrate in Figure S3, such that three separate episodes can be identified on the figure. That is a part of the story that is undersold in the narrative.

I also thought Figure S9 should be in the main body of the paper, since it illustrates another key finding of the modeling work: the mechanism for introducing icebergs into the subtropics during Heinrich events. This is detailed in the last paragraph of the main text, but the only illustrations are in the supplementary material.

Other comments:

Lines 191-194: This sentence is too long and should be broken

Lines 194-195: Change "It is also interesting that" to "In addition,"

Line 201: Delete "it is interesting given that"

Line 206: Delete "Intriguingly,"

Line 217: Delete "remarkable"

Figure 2: the magenta contour lines are difficult to distinguish as distinct from the proximal dark blues of the iceberg density colors (magenta is the next step on the rainbow color scale, and I have a touch of colorblindness). How about black lines, or gray shaded region?

Reviewer #2 (Remarks to the Author):

This is a really neat and well-written paper that takes observations of numerous iceberg scours in shallow waters off of South Carolina and of sediment cores that intercept at least some of the scours to estimate the age(s). They find a significant number of scours that appear to have occurred at ~31 ka, coincident with the timing of Heinrich Event H3. They acknowledge that there are likely many

more events of scours, but emphasize that H3 does stand out as different from the others of the last five events. These observations are really interesting, but what sets the paper apart is the modeling that explains how icebergs may be able to travel down the east coast of the US to get as far south as South Carolina and even the Florida Keys. Using high resolution, eddy resolving modeling, they show that with a significant flux of fresh water (>2.6 Sv) a large number of icebergs follow the freshwater down the coastal current, counter to the Gulf Stream flow. In their models, the greater the fresh water flux the greater the proportion of icebergs make it down to these rather stunning latitudes.

I find the observations and explanations novel, and of broad relevance to our understanding of how iceberg and meltwater discharges may influence hydrographic changes in the North Atlantic and thus may contribute to variations in overturning circulation. I have heard people mention the presence of iceberg scours so far south, but haven't followed it. But I think that this paper with its well displayed observations and convincing numerical simulations will have an important impact on how people think of iceberg rafting from the Laurentide Ice Sheet.

I am an observationalist and I am not able to critically evaluate the numerical simulations that are employed in this paper, but it appears to be well thought out and I am compelled by the results and conclusions. I look forward to seeing this published, and I look forward to follow up studies that will no doubt continue. I would recommend publication as is. Sidney Hemming

Reviewer #3 (Remarks to the Author):

Manuscript review of "The timing of iceberg scours and massive ice-rafting events in the subtropical North Atlantic" by Alan Condron and Jenna Hill

The work presented is a follow-up from a previous publication (Hill and Condron, 2014, *Nature Geoscience*, 7(11), 806-810) by the same tandem of authors. In addition to previous results, in this manuscript, 14C aging of some scours allow them to associate some to Heinrich event 3 (H3). Further high resolution modeling experiments based on iceberg and freshwater forcing lead the authors to propose that the southward coastal routing of iceberg was important and was made possible due to numerous massive meltwater release into the North Atlantic (based on modeling experiments and confirming Hill and Condron 2014 results), justifying also why H3 is recorded by less abundant IRD records in the remaining North Atlantic. They conclude that their results "demonstrate a complex relationship between freshwater forcing and climate change" which is not clearly described in my opinion. Despite the evident interest of their study, it seems to me that their analyses lack to consider major facts about icebergs and H3 that would corroborate their conclusions based on modeling experiments and place their results into a broader perspective. Further discussion of the results is necessary to make this manuscript a valuable contribution for a better knowledge of H3. My concerns and comments are mainly from a paleo perspective since I am not a modeler.

Comments:

H3 and H6 are indeed some of the less documented Heinrich events. However, there is a longtime published work about H3 that would support their scenario and conclusion: Gwiazda et al. 1996 (*Paleoceanography*, vol. 11 (4), pp 371-378) bring evidences that most of the icebergs melted on the western Atlantic, suggesting a large freshwater availability on this side of the Atlantic. H3 also occurred during full-glacial conditions. Why the role of meltwater is not considered since the beginning and only highlighted when a major flux is necessary to bring the iceberg to Florida (5sv experiment)? How a large extension of the Laurentide ice sheet (LIS) would also interfere to this enhanced southward flux of meltwater and icebergs? What about a southward displacement of the Gulf Stream? Would it facilitate the passage of the Hatteras front to the veiling icebergs? The authors conclude about the complex relationship between freshwater forcing and climate without defining it. Is it implicit that the freshwater forcing would be applied to a different location (southern,

coastal?) than usually? However, H3 seems to be the Heinrich event that impacted the AMOC (Henry et al., 2016, *Science*, vol. 353 (6498)) the less, even at Florida Straits (Lynch-Stirglitz et al., 2014, *Nature Geoscience*, vol. 7 pp 144-150). It is necessary that the authors comment that point to clarify their interpretation and clearly highlight the significance of their results.

About their summary, I would suggest the authors to damp a bit their conclusions to be more convincing: hypothesizing that it is because of this coastal routing that there is limited IRD record on the remaining Atlantic during H3, it is forgetting that not all iceberg hold IRD and that the absence of IRD doesn't mean no iceberg. Besides, based on their model, they seem to conclude that coastal routing was the only efficient way to transport icebergs to the subtropics, implicitly recusing the iceberg transport via a structure similar to cold-core rings (in the summary section): still, this coastal routing doesn't cross the Gulf Stream to reach the Sargasso sea.

Comment on the modeling experiments (again, by a non-specialist):

The authors referred to Roberts et al. (2014) to evaluate their freshwater forcing. They should also mention their conclusion that larger icebergs have a greater capacity to travel South since most of the icebergs in the experiment are very large (Table S4: about half of the icebergs). It will justify the iceberg size distribution used in the model.

It will be very interesting that the authors comment on the sensitivity of their model to freshwater forcing and, most importantly, about how the meltwater flux applied is realistic. Although the value of 2.5 Sv is explained in Hill and Condron 2014, it should be also mentioned briefly here. If I understood correctly, it is a unique forcing applied or is it of a longer duration?

Comments along the text:

title: timing of some iceberg scours instead? Heinrich event 3 scours? Not all iceberg scours have been dated and the authors suggest that some may correspond to other Heinrich events.

Line 32: a brief description about the surface hydrography would be welcome (Hatteras front, meeting point of opposite water masses, etc...).

Line 36: "climatic interest" should be explicit.

Line 52: "influence of meltwater [...] more complex": why? cf. discussion about AMOC state during H3 above. It should be also considered that under such extreme cold conditions, meltwater water impact on circulation could have been muted (Lynch-Stieglitz et al 2014 cited above). H3 may not be the best example but it would provide arguments to tackle the issue of meltwater forcing, considering a possible additional routing through the Atlantic.

Lines 64-65: "foraminiferal assemblages": except the later reference to *G. menardii* (considered as a Holocene indicator), there is no data about foraminiferal assemblages.

Lines 90: which is the sampling resolution to look for *G. menardii* (10 cm?)? In every dated core?

Line 130-131: "[...] first time glacial iceberg discharge events have been simulated at such a high spatial resolution". If the authors believe that it is the most important aspect of their manuscript, it should be highlighted since the beginning.

Reviewer #4 (Remarks to the Author):

Using observations of glacial iceberg scouring and an iceberg-enabled ocean model, the authors provide compelling evidence for the remarkable arrival of icebergs in the subtropical North Atlantic, along the SE coast of the US. Model simulations further support an argument that these rare events are only possible under massive (probably brief) freshwater input and associated iceberg calving, from the Laurentide ice sheet.

The iceberg enabled MITgcm model is a state-of-the-art model ideal for the simulations presented here. I agree on the importance of representing iceberg drift as the integral effect of vertical currents (Figure S5), given the strongly sheared flows (evident in Figure S7), and this is likely critical to the results presented here. Another novel feature of the iceberg modelling is to allow scouring to a depth

of 20 m, providing direct comparison with the survey data (Fig. 4c). LGM boundary conditions from a CCSM3 simulation are adequate for the purposes of this study. Details of the survey data and core analysis, along with the model and experiments, are well described in Supplementary Material. In my specific comments below, I request a bit more detail on the model experiments, of relevance to interpretation of the simulations, as well as raising some questions on processes and seasonality.

Specific Comments:

1. While the allowance of scouring in the iceberg dynamics is novel and appropriate, one would expect this to be associated with very strong bottom drag, decelerating the icebergs – this bottom drag does not appear in the (orthodox) iceberg momentum balance, equation (1) of Supplementary Material? Might its exclusion be an issue? Could the authors note this and convince the reader that neglect of bottom drag can be justified, if indeed it is neglected?

2. What are the physical and dynamical mechanisms that support the coastal current as far south as Florida Straits under 5 Sv freshwater input, compared to 2.5 Sv? Under current climate, Slope Water prevails inshore of the Gulf Stream, north of Cape Hatteras – where coastal geometry and bathymetry present a barrier to southward progress of coastal flows (additional to the Gulf Stream). Accepting the rather different coastal geometry and shelf width under lower sea level, it seems that there is a dynamical ‘tipping point’, at which the coastal current can extend southward of the Cape Hatteras ‘barrier’. Noting the extraordinarily low inshore salinity, and strong salinity front, in Fig. S7 (hence baroclinic geostrophic flow via thermal wind - southward, assuming salinity dominates temperature in density terms) – to what extent might inertia of the coastal current (sufficiently strong given a large enough horizontal salinity gradient) be important for ‘overshooting’ Cape Hatteras, while continuing to follow isobaths as a geostrophic slope current (conforming to Taylor-Proudman theorem)? I encourage the authors to explain - with this or alternative reasoning - how a coastal current can extend so far into the subtropics.

3. Table S4: Is the iceberg size distribution that used in present-day ocean-iceberg modelling (e.g. Martin and Adcroft, 2010, Ocean Modelling; Marsh et al. 2015, GMD)? One could argue that the distribution between size classes would be different at Heinrich Events, perhaps proportionately more icebergs in the large size classes? This is perhaps worth some reflection (in SM), along with justification for the size distribution used here.

4. Given the rapid transit from Hudson Bay to the subtropics (0.25-1 years, Fig. S8), and the melting scheme in equations (6)-(12), how is the subtropical survival of icebergs related to size distribution and seasonality (hinted in Fig. S8)? Can the authors relate not just the presence of icebergs in the subtropics to an advective route, but also the relatively limited melt rates in a presumably cold coastal current; only salinity is shown in the figures, apart from Jan/Sept SST in the control (Fig. S2), but I assume temperatures are well below those of offshore waters.

5. Of relevance to the previous point, at what time of year and for how long are the 2.5 Sv or 5 Sv floods imposed on the model? Assuming these floods coincide with the iceberg fluxes (0.2 Sv sustained for 1 year?), then one would assume that the freshwater and iceberg fluxes are applied continuously for 1 year. Given transit times, and substantial seasonality in ocean temperature (Fig. S2), quite different model results may be obtained for floods and iceberg fluxes constrained to different seasons.

6. It seems that a key outcome is the indication that only truly massive floods – but perhaps very short-lived – can explain scouring that extends so far south. Shorter duration, more intense floods (than previously supposed) could be more clearly emphasized in Abstract and Summary.

Reviewer: Professor Robert Marsh

Reviewer 1:

Comment 1: I do hope the authors can expand their figures shown in the main body of the paper. Figure 1 is, unfortunately, a particularly poor figure because too much was crammed into it, such that much of the important information and annotation were unreadable without significant magnification of the image. It is really 3 figures – panels A-D, panel E, and panel F. And instead of Figure 1E, I would hope the authors could move supplementary figure S3 into the main body of the text. Figure S3 is an excellent figure that best highlights the key findings of the sedimentary component of the paper. The chirp data are particularly important in confirming the existence of multiple episodes of iceberg scouring, even if all of them cannot be clearly dated. But that information, combined with the modeling work, provides a very compelling rationale for concluding that these must have been Heinrich events, because otherwise there is no way to get icebergs to this location. I was surprised, for example, that the authors did not make note in the text of the scour identified in the hard-bottom substrate in Figure S3, such that three separate episodes can be identified on the figure.

Reply 1: We entirely agree with you about Figure 1 and have now split this into several figures as well as moved some of the related figures in the Supplementary Section into the main document. They are now as follows: Figure 1 comprises of panels A-D from the original; Figure 3 shows the chirp subbottom profile across the depocenter where core GC27 was collected; Figure 4 shows just the radiocarbon dates from sediment cores collected in and around buried iceberg scours; and Figure 5 shows Core GC24, collected in a deeper portion of the depocenter, where no iceberg scours are observed. We have also made explicit note in the text to the presence of the iceberg scour identified in the hard-bottom substrate in Figure 3 at ~238cm depth.

Comment 2: I also thought Figure S9 should be in the main body of the paper, since it illustrates another key finding of the modeling work: the mechanism for introducing icebergs into the subtropics during Heinrich events. This is detailed in the last paragraph of the main text, but the only illustrations are in the supplementary material.

Reply 2: Indeed. As suggested, Figure S9 has now been moved to the main manuscript and is now Figure 10.

Comments 3 & 4: Lines 191-194: This sentence is too long and should be broken. Also, lines 194-195: Change “It is also interesting that” to “In addition,”

Reply 3-4: This paragraph has now been reworded to say:

“However, after one year of elevated meltwater discharge the geographical distribution of icebergs has significantly expanded to include much of the subtropical western North Atlantic, such that icebergs drift as far south as Bermuda Rise. Once the meltwater discharge is reduced though, icebergs become restricted to the subpolar North Atlantic even though the freshwater signature of the meltwater persists in the subtropics (Fig. 10c).

Comments 5-7:

Line 206: Delete “Intriguingly,”

Line 217: Delete “remarkable”

Line 201: Delete “it is interesting given that”

Reply 5-7: These three suggestions were all implemented.

Comment 8:

Figure 2: the magenta contour lines are difficult to distinguish as distinct from the proximal dark blues of the iceberg density colors (magenta is the next step on the rainbow color scale, and I have a touch of colorblindness). How about black lines, or gray shaded region?

Reply 8: Thank you for catching this. We re-made the figure using a gray line instead of magenta.

Reviewer 2

“I look forward to seeing this published, and I look forward to follow up studies that will no doubt continue. I would recommend publication as is”

Reply: Thank you for your support! We also look forward to future discussions with you on this topic.

Alan Condron & Jenna Hill

Reviewer 3:

Comment 1:

H3 and H6 are indeed some of the less documented Heinrich events. However, there is a longtime published work about H3 that would support their scenario and conclusion: Gwiazda et al. 1996 (*Paleoceanography*, vol. 11 (4), pp 371-378) bring evidences that most of the icebergs melted on the western Atlantic, suggesting a large freshwater availability on this side of the Atlantic. H3 also occurred during full-glacial conditions. Why the role of meltwater is not considered since the beginning and only highlighted when a major flux is necessary to bring the iceberg to Florida (5sv experiment)?

Reply 1:

You raise a very important point about prior research on H3 & H6 regarding the possible confinement of icebergs to the western North Atlantic. Thank you. We're now added the following text to note this:

“Previous work indicates that H3 has several features that set this event apart from four of the six Heinrich layers (H1, H2, H4, and H5) that occurred in the last 60 kyr (*e.g., ref. 14, 25, 26*). In particular, while IRD from Hudson Bay is found in the western North Atlantic during H3, it is significantly lacking in the eastern North Atlantic sector compared to these four H-events. In fact, IRD in the eastern North Atlantic appears to have been sourced from the Greenland and/or the Eurasian ice sheets during H3 (*14, 25*). Heinrich Event 6 also shows a similarly modest increase in IRD and possibly a different IRD source, compared to the other 4 major events (*14, 26*). Gwiazda et al., (*ref. 26*) proposed that this variation in IRD deposition during H3 reflected a greater confinement of icebergs sourced from the LIS to the western North Atlantic, but precisely why this might have been the case remains unknown. Here, we postulate that the repeat transport of icebergs to the western subtropical North Atlantic by large meltwater floods could explain this pattern, especially if such events occurred multiple times during H3.”

New references:

Ref 25: Grousset, F.E., Labeyrie, L., Sinko, J.A., Cremer, M., Bond, G., Duprat, J., Cortijo, E. and Huon, S., 1993. Patterns of ice-rafted detritus in the glacial North Atlantic (40–55° N). *Paleoceanography*, 8(2), pp.175-192.

Ref 26: Gwiazda, R.H., Hemming, S.R. and Broecker, W.S., 1996. Provenance of icebergs during Heinrich event 3 and the contrast to their sources during other Heinrich episodes. *Paleoceanography*, 11(4), pp.371-378.

To your second point about ‘why the role of meltwater is not considered since the beginning [of the manuscript]’: This is because our work focuses on explaining the mechanism(s) responsible for the transport of icebergs to exceptionally warm, subtropical, regions of the western North Atlantic where the Gulf Stream flows northward and opposes any southerly iceberg drift. In all of our model runs (to-date), only truly massive (but short-lived) meltwater floods are capable to

transporting icebergs to this region. As the AMS dating of the scours suggests these features were formed during H3 we conclude that there must have been times during this period when enormous outburst floods occurred. The main focus of our manuscript has now been more clearly stated on line 64:

“In this manuscript, we report on the sedimentology and ages of several buried iceberg scour marks observed on the subtropical U.S. continental margin, south of Cape Hatteras. We then use an iceberg model, coupled to a high-resolution (eddy permitting) ocean-sea ice model, to determine the mechanisms that led to the formation of these features. Finally, we conclude by considering the implications of our results for understanding the factors controlling the patterns of ice-rafted debris (IRD) across the subpolar North Atlantic (i.e., the IRD-belt) and the role that meltwater input to the ocean plays in modulating deep-water formation and large-scale ocean circulation.”

Comment 2:

How a large extension of the Laurentide ice sheet (LIS) would also interfere to this enhanced southward flux of meltwater and icebergs?

Reply 2:

In our study we mainly focus on Hudson Strait as the source of the meltwater and icebergs given past research on the source of icebergs from the Laurentide Ice Sheet (LIS) during Heinrich Events. You do, however, raise a good point about the role of icebergs and meltwater from more southerly locations. In Hill and Condron (2014, Nature Geoscience) we did release meltwater from the Gulf of St. Lawrence and found broadly similar results in terms of the meltwater pathways along the east coast of North America and penetration of the meltwater south of Cape Hatteras. As such, any icebergs calved from the LIS south of Hudson Bay that were in the immediate pathway of the meltwater would have carried along with the meltwater and rapidly transported southwards towards the subtropical scour sites. It is also possible that icebergs originating from more far-field calving margins such as Greenland and Iceland could have reached these sites as well if they in the pathway of the meltwater. We now comment on these points on line 230:

“In addition, while our experiments only consider the transport of icebergs calved from the Hudson Bay region, it is entirely possible that icebergs originating from more southerly parts of the LIS and/or far-field calving margins such as Greenland and Iceland could have been ‘swept along’ with the meltwater (provided they were south of the drainage outlet at the time of the flood) and contributed to the formation of the subtropical iceberg plow marks.”

Comment 3:

What about a southward displacement of the Gulf Stream? Would it facilitate the passage of the Hatteras front to the veiling icebergs?

Reply 3:

Thank you for asking this very interesting question. We actually performed this experiment but it was never included in the final manuscript. The result (shown below) is that icebergs can now reach the most northern relic scour sites off the coast of South Carolina, and this is also where the greatest number of plow marks are been identified on the sea floor. Despite this, however, icebergs were still unable to freely drift to the scour sites located further south, i.e., those directly beneath the Gulf Stream. Given the relevance of this finding, we've included the figure below in the manuscript (**Fig. S3**) and added the following text to lines 163-176:

“To explore the relationship between the northward flow of the Gulf Stream and the southward flow of the slope water in controlling the southern limit of iceberg drift, an additional model experiment was performed with the wind field over the North Atlantic shifted south to artificially the push the Gulf Stream south (**See Methods**). In response to this change in wind forcing, the Gulf Stream detached $\sim 1^\circ$ further south of Cape Hatteras, compared to the control simulation, and allowed icebergs to freely drift to the most northern relict scour sites off the coast of South Carolina. Significantly, this is also where the greatest number of plow marks have previously been identified on the sea floor (*ref. 1*), suggesting that precise latitude at which the Gulf Stream separates from the coast in a glacial ocean (*21*) controls the ability of icebergs to reach the most northern scour location. South of this region, however, the persistent northward flow of the Gulf Stream continued to inhibit icebergs from reaching the scour sites located off the coast of Florida (**Fig. S3**). A different forcing mechanism - rather than a change in the position and/or detachment of the Gulf Stream from the coast - is thus needed to explain the occurrence of most of the scour features.

New Reference:

Ref 21: Matsumoto, K. and Lynch-Stieglitz, J., 2003. Persistence of Gulf Stream separation during the Last Glacial Period: Implications for current separation theories. *Journal of Geophysical Research: Oceans*, 108(C6).

Figure S3: The simulated distribution of icebergs in the glacial North Atlantic in response to a southward shift in the latitude of the Gulf Stream. Compared to the Control simulation, a small number of icebergs drift to the most northern relic scour sites - located off the coast of South Carolina, USA - due to slope waters now flowing further south at Cape Hatteras. Icebergs were nevertheless still unable to reach the most southerly scour sites located off the coast of Florida that are directly beneath the northward flowing Gulf Stream. For reference, the Gulf Stream is marked by the 13-15°C isotherms at 200m water depth (grey contour lines). Iceberg calving margins near Hudson Bay are denoted by the white arrows, glacial landmasses are shown in black.

A further point to make here (and one that is very relevant to your first question) is that this shift in the Gulf Stream also results in a greater confinement of icebergs to the western North Atlantic, and less icebergs drifting eastwards to Europe. It is therefore possible that a southward displacement of the Gulf Stream might also explain the lack of Hudson-derived IRD in the eastern North Atlantic during H3. This is now noted on line 281:

“We also note that a more southerly position of the Gulf Stream could have, in part, contributed to the observed change in IRD during H3 given that our model predicts an increased confinement of icebergs to the western North Atlantic when the wind field was perturbed (Fig. S3).”

Comment 4:

The authors conclude about the complex relationship between freshwater forcing and climate without defining it. Is it implicit that the freshwater forcing would be applied to a different location (southern, coastal?) than usually? However, H3 seems to be the Heinrich event that impacted the AMOC (Henry et al., 2016, *Science*, vol. 353 (6498)) the less, even at Florida Straits (Lynch-Stirglitz et al., 2014, *Nature Geoscience*, vol. 7 pp 144-150). It is necessary that the authors comment that point to clarify their interpretation and clearly highlight the significance of their results.

Reply 4:

Again, thank you for raising this point as its relevant to the extent to which freshwater forcing from H3 likely impacted AMOC strength. In short, massive meltwater outburst floods from Hudson Bay, Canada, end up being transported directly to the subtropical North Atlantic gyre, thousands of kilometers south of regions of deep-water formation. This finding was previously noted in Condrón and Winsor (2012) and Hill and Condrón (2014). To return to this point in light of our latest results showing that outburst events occurred during H3, we have additional text (line 305+). In short, we propose that the subtropical transport iceberg-meltwater pathway we identify could have caused a more muted AMOC response during H3 given that freshwater is initially routed thousands south of regions of NADW formation that regulate AMOC strength:

“Our findings thus demonstrate that the geographical region of the ocean influenced by meltwater freshening was not confined to the subpolar gyre but is consistent with previous studies (1, 6) showing that the release of large volumes of iceberg-laden meltwater from Hudson Bay, Canada, leads to a significant freshening of the subtropical North Atlantic gyre (**Fig. 10**). This freshwater then undergoes significant mixing and is gradually advected northwards by the Gulf Stream towards the subpolar gyre. As a result, the freshwater is much saltier (less fresh) by the time it reaches high-latitude regions of deep-water formation (that likely modulate AMOC strength) than if it had been directly released to the subpolar gyre. This result is in contrast to both the notion that subpolar regions of deep-water formation were rapidly freshened by large outburst floods and the ‘classic’ technique in numerical models of applying a uniform layer of freshwater to the subpolar North Atlantic (between 50-70°N) to study the impact of freshwater on AMOC and climate (*ref. 4, 5*). We postulate that the initial transport of significant volumes of freshwater to the subtropical North Atlantic as a result of massive glacial outburst floods, followed by the subsequent mixing of this water with the ambient ocean *en-route* to the subpolar gyre, could explain the muted reduction in AMOC strength during Heinrich Event 3 (28) given that meltwater would be saltier by the time it reached the subpolar gyre, and thus less capable of inhibiting deep-water formation.”

New Reference:

Ref 28: Henry, L.G., McManus, J.F., Curry, W.B., Roberts, N.L., Piotrowski, A.M. and Keigwin, L.D., 2016. North Atlantic ocean circulation and abrupt climate change during the last glaciation. *Science*, 353(6298), pp.470-474

Comment 5:

About their summary, I would suggest the authors to damp a bit their conclusions to be more convincing: hypothesizing that it is because of this coastal routing that there is limited IRD record on the remaining Atlantic during H3, it is forgetting that not all iceberg hold IRD and that the absence of IRD doesn't mean no iceberg.

Reply 5:

To address this point, we have now added the following text to lines 284+

“Given uncertainties in the concentration and partitioning of IRD within glacial icebergs (e.g., 27), we also cannot rule out the possibility that a lack of IRD deposition during Heinrich Event 3 simply reflects a change in the concentration of IRD in the icebergs and/or a change in where the IRD is partitioned within the ice at this time. Indeed, ‘clean’ icebergs with little or no IRD - analogous to modern-day icebergs calved from large ice shelves fringing Antarctic - would leave little or no IRD ‘fingerprint’ on the sea floor, while icebergs with IRD concentrated in the basal portion of the ice would cause IRD to be deposited much closer to the calving margin.

New reference

Ref 27: Andrews, J.T., 2000. Icebergs and iceberg rafted detritus (IRD) in the North Atlantic: facts and assumptions. *Oceanography*, pp.100-108.

Comment 6:

Besides, based on their model, they seem to conclude that coastal routing was the only efficient way to transport icebergs to the subtropics, implicitly recusing the iceberg transport via a structure similar to cold-core rings (in the summary section): still, this coastal routing doesn't cross the Gulf Stream to reach the Sargasso Sea.

Reply 6:

In the conclusions section we acknowledge that the coastal meltwater pathway we simulate in our model offers an alternative mechanism to cold-core rings as a way to transport meltwater offshore into the subtropical gyre (Sargasso Sea). You are incorrect, however, in stating that the coastal meltwater routing doesn't cross the Gulf Stream and reach the Sargasso Sea. To more clearly illustrate this, we've now included an additional figure (Figure 10) showing snapshots of sea surface salinity and icebergs in the North Atlantic at the onset of the meltwater event, after one year of elevated meltwater discharge, and after the meltwater flood has ended. In this figure it is clear that both freshwater and icebergs are transported offshore into the Sargasso Sea.

Figure 10: Snapshots of sea surface salinity and the distributions of icebergs in the North Atlantic. The top panel (A) shows that at the onset of the meltwater event, icebergs are primarily restricted to the region 40°N-50°N, where high concentrations of IRD are found in marine sediments. After one year of elevated meltwater discharge the geographical distribution of icebergs has expanded to include the subtropical North Atlantic. Once the meltwater discharge is reduced (panel C), the geographical distribution of icebergs ocean again becomes restricted to the subpolar North Atlantic even though the freshwater signature of the meltwater persists in the subtropics. Note that icebergs are represented by the white triangles.

Comment 7:

Comment on the modeling experiments (again, by a non-specialist): The authors referred to Roberts et al. (2014) to evaluate their freshwater forcing. They should also mention their conclusion that larger icebergs have a greater capacity to travel South since most of the icebergs in the experiment are very large (Table S4: about half of the icebergs). It will justify the iceberg size distribution used in the model.

Reply 7:

The iceberg size distribution we use approximates present-day iceberg sizes (now noted in the Methods Section) and in fact the majority (66%) of these icebergs are fairly small (i.e. ≤ 267 m wide x 320 m thick). Given uncertainties in the size of icebergs associated with Heinrich Events we consider this to be a reasonable first approximation especially as iceberg scouring in our model occurs in the same water depths as the observations (Figure 9).

Comment 8:

It will be very interesting that the authors comment on the sensitivity of their model to freshwater forcing and, most importantly, about how the meltwater flux applied is realistic. Although the value of 2.5 Sv is explained in Hill and Condron 2014, it should be also mentioned briefly here. If I understood correctly, it is a unique forcing applied or is it of a longer duration?

Reply 8:

In our meltwater simulations, freshwater fluxes of 2.5Sv and 5Sv were released (in separate experiments) from Hudson Bay for 1 year to simulate the rapid drainage of a large proglacial lake to a new level. Reconstructions of the volumes of freshwater released to the ocean during these outburst events are poorly known, but they are estimated to have peaked at 5 Sv during the 8.2 kyr event (Barber et al., 1999). The time taken for a lake to lower to its new outlet is also uncertain, although hydrologic modelling estimates suggest that these events may have lasted only for up to 1 year (Clarke et al., 2004). In addition, a 0.2 Sv flux of icebergs from Hudson Bay was applied constantly throughout our model simulations, both prior to the release of any meltwater, during the meltwater experiments, and after the meltwater flood ceased. We have now noted these points beginning on Line 177-190.

New references

Ref 22: Barber, D. C. et al. Forcing of the cold event of 8,200 years ago by catastrophic drainage of Laurentide lakes. *Nature* 400, 344-348 (1999).

Ref 23: Clarke, G. K. C., Leverington, D.W., Teller, J. T. & Dyke, A. S. Paleohydraulics of the last outburst flood from glacial Lake Agassiz and the 8200BP cold event. *Quat. Sci. Rev.* 23, 389-407 (2004).

Comments along the text:

title: timing of some iceberg scours instead? Heinrich event 3 scours? Not all iceberg scours have been dated and the authors suggest that some may correspond to other Heinrich events.

Reply: Given that there are still many un-dated iceberg scours in our study region we have decided to keep the current title.

Line 32: a brief description about the surface hydrography would be welcome (Hatteras front, meeting point of opposite water masses, etc...).

Reply: We have now included the following additional description of the surface hydrography on lines 31-35:

“Indeed, in our prior work (1) the Gulf Stream in the glacial North Atlantic flows north along the continental shelf of North America until it detaches from the coast near Cape Hatteras, much like present day (2). In the Mid-Atlantic Bight region to the north, cold subpolar slope waters flow south from the Grand Banks of Newfoundland until they encounter the Gulf Stream at Cape Hatteras (**Fig. 2**). Hence, for icebergs to reach the subtropical scour locations south of Cape Hatteras they must have drifted against the normal northward direction of flow over the continental; i.e., in the opposite direction to the Gulf Stream.”

New reference

Ref 2: Pietrafesa, L.J., Morrison, J.M., McCann, M.P., Churchill, J., Böhm, E. and Houghton, R.W., 1994. Water mass linkages between the Middle and South Atlantic Bights. *Deep Sea Research Part II: Topical Studies in Oceanography*, 41(2-3), pp.365-389

Line 36: “climatic interest” should be explicit.

Reply: Indeed. We have now reworded this to say “for understanding cryosphere-ocean-climate interactions.”

Line 52: “influence of meltwater [...] more complex”: why? cf. discussion about AMOC state during H3 above. It should be also considered that under such extreme cold conditions, meltwater water impact on circulation could have been muted (Lynch-Stieglitz et al 2014 cited above). H3 may not be the best example but it would provide arguments to tackle the issue of meltwater forcing, considering a possible additional routing through the Atlantic.

Reply: We agree. Please see our reply to Comment 4 (above)

Lines 64-65: “foraminiferal assemblages”: except the later reference to *G. menardii* (considered as a Holocene indicator), there is no data about foraminiferal assemblages.

Reply: We have removed this phrase from the text. We did some initial work looking at foraminiferal assemblages, but are not reporting that here as it turned out to be more peripheral.

Lines 90: which is the sampling resolution to look for *G. menardii* (10 cm?)? In every dated core?

Reply: All of the dated cores were sampled at 10cm resolution to look for *G. menardii*, but none were observed.

Line 130-131: “[...] first time glacial iceberg discharge events have been simulated at such a high spatial resolution”. If the authors believe that it is the most important aspect of their manuscript, it should be highlighted since the beginning.

Reply: Agree. We have now moved this sentence to Line 127 and also noted it in the abstract.

Reviewer 4

Thank you for the many interesting and inciteful comments regarding the modeling aspect of our work. Your points are all well taken and have helped us develop our manuscript further. Indeed, once our article is published, we look forward to additional discussions (and possible collaborations) given the obvious overlaps in our research.

Comment 1: While the allowance of scouring in the iceberg dynamics is novel and appropriate, one would expect this to be associated with very strong bottom drag, decelerating the icebergs – this bottom drag does not appear in the (orthodox) iceberg momentum balance, equation (1) of Supplementary Material? Might its exclusion be an issue? Could the authors note this and convince the reader that neglect of bottom drag can be justified, if indeed it is neglected?

Reply 1: A full model accounting for the bottom drag caused by icebergs plowing the sea floor was considered too complicated at this stage given it would need to account for both the rheology of the marine sediment and the precise shape of the iceberg keel below the water line. Bottom drag is, however, considered in the model in a simplified form as we assume that an iceberg keel can plow through the marine sediment up to a thickness of 20m, below which the iceberg becomes grounded and stationary (i.e., velocity is set to zero). The iceberg then remains stationary until it sufficiently melts and begins drifting again. We consider this to be a good first approximation given that most of the observed scours are incised up to 20 meters deep into the sea floor sediment. This has now been noted on lines 145+.

Comment 2: What are the physical and dynamical mechanisms that support the coastal current as far south as Florida Straits under 5 Sv freshwater input, compared to 2.5 Sv? Under current climate, Slope Water prevails inshore of the Gulf Stream, north of Cape Hatteras – where coastal geometry and bathymetry present a barrier to southward progress of coastal flows (additional to the Gulf Stream). Accepting the rather different coastal geometry and shelf width under lower sea level, it seems that there is a dynamical ‘tipping point’, at which the coastal current can extend southward of the Cape Hatteras ‘barrier’. Noting the extraordinarily low inshore salinity, and strong salinity front, in Fig. S7 (hence baroclinic geostrophic flow via thermal wind - southward, assuming salinity dominates temperature in density terms) – to what extent might inertia of the coastal current (sufficiently strong given a large enough horizontal salinity gradient) be important for ‘overshooting’ Cape Hatteras, while continuing to follow isobaths as a geostrophic slope current (conforming to Taylor-Proudman theorem)? I encourage the authors to explain - with this or alternative reasoning - how a coastal current can extend so far into the subtropics.

Reply 2:

To address your interesting point about the ability of a meltwater flood to continue south at Cape Hatteras, i.e., to ‘overshoot’, we have now produced snapshots of sea surface height (SSH) anomalies in the western subtropical North Atlantic for simulations releasing 1 Sv, 2.5 Sv, and 5 Sv of meltwater from Hudson Bay (See below). This figure has also been included in the Supplementary material as Figure S4.

The meltwater current is essentially a semi-geostrophic buoyant gravity current that is observable in the model as a ‘bulge’ in the sea surface height (SSH). Consistent with theoretical and laboratory studies of buoyant gravity currents along a sloping bottom in a rotating fluid (ref Lentz and Helfrich, 2002), the vertical thickness of the meltwater is influenced by the original magnitude of the meltwater flood, with larger discharge events producing buoyant gravity currents that are thicker and also extend farther offshore. The ability of the meltwater to continue to flow south at Cape Hatteras thus depends on whether the SSH of the meltwater is larger than the sea surface height associated with the Gulf Stream, which in the model is found to be the case for both the 2.5Sv and 5Sv outburst floods, but not the 1 Sv flood. Indeed, in the 1 Sv flux experiment, both icebergs and meltwater are retroflected eastward into the interior of the subtropical Atlantic gyre. In the 2.5 Sv experiment, although the meltwater penetrates as far south as Florida Strait, icebergs cannot reach the most southern scour sites due to the meltwater becoming increasingly shallow (~20m thick at Florida Strait) to the extent that the drag force exerted on the lower part of the icebergs by the northward flowing Gulf Stream exceeds the southward surface drag from the meltwater. It is only in the 5 Sv experiment that both meltwater and icebergs reach the most southerly scour sites. The above points have now been mentioned on lines 196-207.

New reference

Ref 24: Lentz, S. J. & Helfrich, K. R. Buoyant gravity currents along a sloping bottom in a rotating fluid. *J. Fluid Mech.* 464, 251_278 (2002)

Figure S4: Change in sea surface height in the subtropical western North Atlantic in response to elevated meltwater forcing from Hudson Bay, Canada. The panels (a-c) show the change in sea surface height (Perturbation minus Control) resulting from a 1 Sv, 2.5 Sv, and 5 Sv meltwater flood. The ability of the meltwater to flow south at Cape Hatteras, i.e. to ‘overshoot’, is dependant on whether the height of the meltwater exceeds the ambient sea surface height. This is the case for both the 2.5Sv and 5Sv meltwater floods, but not the 1Sv flood.

Comment 3: Table S4: Is the iceberg size distribution that used in present-day ocean-iceberg modelling (e.g. Martin and Adcroft, 2010, Ocean Modelling; Marsh et al. 2015, GMD)? One could argue that the distribution between size classes would be different at Heinrich Events, perhaps proportionately more icebergs in the large size classes? This is perhaps worth some reflection (in SM), along with justification for the size distribution used here.

Reply 3: The iceberg size distribution is the same as in Bigg et al. (1997) and approximates present-day iceberg sizes. Given uncertainties in the size of icebergs associated with Heinrich Events we consider this to be a reasonable estimate. Additionally, Figure 9 shows that iceberg scouring in our model occurs in roughly the same water depths as the observations which suggests that the sizes of the icebergs we are using fairly accurately reflects the size of those in the past. We have now noted this in the Methods Section.

Ref. 33: Bigg, G.R., Wadley, M.R., Stevens, D.P. and Johnson, J.A., 1997. Modelling the dynamics and thermodynamics of icebergs. *Cold Regions Science and Technology*, 26(2), pp.113-135.

Comment 4: Given the rapid transit from Hudson Bay to the subtropics (0.25-1 years, Fig. S8), and the melting scheme in equations (6)-(12), how is the subtropical survival of icebergs related to size distribution and seasonality (hinted in Fig. S8)? Can the authors relate not just the presence of icebergs in the subtropics to an advective route, but also the relatively limited melt rates in a presumably cold coastal current; only salinity is shown in the figures, apart from Jan/Sept SST in the control (Fig. S2), but I assume temperatures are well below those of offshore waters.

Reply 4: This is a great point and to address it we've added an additional figure to our manuscript (Fig. 8) that shows the change in sea surface temperature in the subtropical western North Atlantic in response to elevated meltwater discharge (shown below): The snapshots (a-c) are drawn 60, 75 and 90 days after the 5 Sv meltwater flood was released from Hudson Bay, Canada, and correspond to the same periods as panels e-g in Figure 7 in the main manuscript. Panel d also shows how the sea surface temperatures at Florida Strait change over time for the 2.5 Sv and 5 Sv flood. The figure shows that the coastal meltwater current is $\sim 5-8^{\circ}\text{C}$ when it flows south through Florida strait, and thus significantly colder than the $\sim 20-25^{\circ}\text{C}$ waters offshore. The persistence of this cold meltwater current significantly reduces ice melt as the icebergs move out of the cold subpolar region into the tropical western North Atlantic. This result is now noted on lines 220-225.

Comment 5: Of relevance to the previous point, at what time of year and for how long are the 2.5 Sv or 5 Sv floods imposed on the model? Assuming these floods coincide with the iceberg fluxes (0.2 Sv sustained for 1 year?), then one would assume that the freshwater and iceberg fluxes are applied continuously for 1 year. Given transit times, and substantial seasonality in ocean temperature (Fig. S2), quite different model results may be obtained for floods and iceberg fluxes constrained to different seasons.

Reply 5: In all of our simulations, both meltwater floods were released for 1 year starting from January 01. This is now more clearly noted in our manuscript on line 183. The 0.2Sv iceberg discharge is actually constant throughout the simulations, such that icebergs are calved at this rate both before, during, and after the meltwater floods begin and end. The flood experiments are therefore starting from a ‘spun-up’ state where icebergs are already drifting across the IRD belt, and as such there is no issue with the timing of the meltwater flood aligning with the period when icebergs are being calved.

Your point about the season of the onset of the flood is interesting. In our experiment with a sustained 1-year 5Sv meltwater flood from Hudson Bay there is an initial pulse of icebergs to Florida, a period when the meltwater current persists, but icebergs are no longer transported south of Cape Hatteras, and then a second period when iceberg transport to the sites of the most southern scours resumes. The period with limited iceberg transport to Florida coincides with the summer months (June – Aug) when slope waters do not penetrate as far south of Cape Hatteras. Considering this, it is possible that the exact timing of the flood would impact the time of arrival of icebergs off the coast of Florida, but given the estimated one-year duration of these events, a flood occurring in the middle of summer would still lead to icebergs reaching Florida, but simply delayed by a few months until the slope waters penetrated further south again.

There are also two final comment related to this point to make here: The first is that the timing (in the geologic record) of when these outburst floods occurred is still very poorly resolved, and secondly, that our future research is planning to develop higher resolution models to further refine the timing, magnitude, and duration of the outburst floods needed to create these remarkable features off the coast of Florida.

Comment 6: It seems that a key outcome is the indication that only truly massive floods – but perhaps very short-lived – can explain scouring that extends so far south. Shorter duration, more intense floods (than previously supposed) could be more clearly emphasized in Abstract and Summary.

Reply 6: Indeed! All of our results point to the fact that the iceberg plow marks off the coast of Florida are direct evidence of massive, but short-lived floods during the last glacial cycle. We have now mentioned this point in the abstract.

REVIEWERS' COMMENTS:

Reviewer #1 (Remarks to the Author):

The authors have done a fine job in revision and have satisfactorily addressed my comments. I think this will be a very impactful paper and I look forward to seeing it in print.

I did notice a couple of issues on Figure 3. Panel B is not described in the caption, and the grain size plot needs a color bar.

John Goff

Reviewer #3 (Remarks to the Author):

I am very pleased that the authors carefully considered the comments of the reviewers. I am satisfied by their precise answers to my questions: the manuscript presents more results and details, strengthening their interpretation of one of the less documented HE. I hope the manuscript will be published very soon since the value of their work is unquestionable for paleoceanographers.

Reviewer #4 (Remarks to the Author):

The authors have clearly responded to my comments and those of Reviewers 1 and 3. In relation to my six specific comments, clarifications and revisions are appreciated:

1. The drag associated with scouring is represented to first order
2. The dynamics of a cold boundary current beyond Cape Hatteras is now clear, and the sea surface height perspective (supported by Fig. S4) is welcome
3. The use of a 'modern' iceberg size distribution is justified
4. 'Survival' to subtropical latitudes of a few large icebergs, due to rapid transit in very cold coastal boundary current, is confirmed with Fig. 8 and accompanying text
5. The issue of flood timing and the background (constant) iceberg calving is resolved and understood
6. The inference of a short duration and massive flood is now clear

My recommendation is that the manuscript is now acceptable for publication in Nature Communications.